# Cytokine concentrations in people with eating disorders: A comprehensive updated systematic review and meta-analysis

Johanna L. Keeler ◉ [1] ✉, Charlotte Bovenberg[2], Hubertus Himmerich ◉ [1,3], Janet Treasure[1,3], Ben Carter ◉ [4], Ulrike Schmidt[1,3] & Bethan Dalton[3,5]

## Abstract

**Background:** Prior research has found altered levels of cytokines in people with eating disorders (EDs). This study is an update of a previous meta-analysis, including longitudinal analyses and machine learning heterogeneity analyses (MetaForest).
**Methods:** This pre-registered (https://osf.io/g6d3f) systematic review and meta-analysis following PRISMA guidelines assessed studies from four databases (PubMed, Web of Science, MEDLINE, PsycINFO) reporting cytokine concentrations in people with EDs until 10th November 2024. Random-effects models were utilised for all meta-analyses.
**Results:** Twenty-four new studies are incorporated, resulting in a total of 43 studies included in meta-analyses. Interleukin (IL)−6 and IL-15 are higher, and IL-7 lower, in anorexia nervosa (AN) compared with controls. When controlling for outliers, tumour necrosis factor (TNF)-α, IL-1β, IL-4, IL-8, IL-10, interferon (IFN)-γ, monocyte chemoattractant protein (MCP) and transforming growth factor (TGF)-β are similar between AN and controls. Longitudinally, IL-6 is lower in AN at follow-up compared to baseline, although this may be an artefact of publication bias. TNF-α and IL-1β do not change longitudinally. There are largely no differences in IL-6 and TNF-α in bulimia nervosa (BN) and there are insufficient studies to perform meta-analyses for binge eating disorder or other EDs.
**Conclusions:** In acute AN, concentrations of IL-6 and IL-15 are elevated and IL-7 is decreased, with preliminary but unconclusive evidence for small decreases in IL-6 over the course of weight restoration. Other cytokines considered to have broadly pro-inflammatory functions are not increased in AN. In BN, there is little evidence for increases in pro-inflammatory cytokines, but the evidence base is limited.

## Plain language summary

This study looked at how certain immune system markers, called cytokines, are affected in people with eating disorders (EDs). Previous research suggested that people with EDs might have altered levels of these markers. In this study, 43 studies were analysed to compare cytokine levels in people with anorexia nervosa (AN), bulimia nervosa (BN), and other EDs. It was found that people with AN had higher levels of certain cytokines (IL-6 and IL-15) and lower levels of others (IL-7), when compared to healthy people. For bulimia nervosa, there was little evidence of altered cytokine levels. These findings are important because they may help improve understanding of the immune system's role in EDs and suggest areas for future research.

Eating disorders (EDs) including anorexia nervosa (AN), bulimia nervosa (BN), binge eating disorder (BED), avoidant/restrictive food intake disorder (ARFID) and other specified feeding and eating disorder (OSFED), involve alterations in feeding and eating patterns and in some cases compensatory behaviours as an attempt to prevent weight gain or offset calorie intake. This group of disorders results in secondary consequences that impact physical systems within the body. One of these is the immune system; both the food restriction seen in EDs such as AN and patterns of overeating seen in binge-type EDs may lead to changes in the production and functioning of immune cells. Additionally, recent

[1]Centre for Research in Eating and Weight Disorders, Department of Psychological Medicine, Institute of Psychiatry, Psychology and Neuroscience, King's College London, London, UK. [2]Department of Psychosis Studies, Institute of Psychiatry, Psychology and Neuroscience, King's College London, London, UK. [3]South London and Maudsley NHS Foundation Trust, Bethlem Royal Hospital, Monks Orchard Road, Beckenham, Kent, UK. [4]Department of Biostatistics and Health Informatics, Institute of Psychiatry, Psychology and Neuroscience, King's College London, London, UK. [5]Department of Psychology, Institute of Psychiatry, Psychology and Neuroscience, King's College London, London, UK. ✉e-mail: johanna.keeler@kcl.ac.uk

research has implicated the presence of autoimmune conditions as a risk factor for the development of an ED, and vice versa[1].

Prior research has examined peripheral concentrations of cytokines in ED populations as indicators of changes in immune signalling. Cytokines are signalling proteins produced by immune cells, endothelial and epithelial cells, adipocytes and connective tissue, which can aid communication between cells in order to modulate the immune response. For example, by stimulating or slowing down the immune system or stimulating the movement of cells toward sites of infection or inflammation. They can have systemic as well as local effects and are pleiotropic in that a single cytokine can produce multiple biological functions. Common groups of cytokines include chemokines, which induce chemotaxis (cell migration); interleukins (IL), which act as chemical signals between white cells; interferons, which help the body resist viral infections and cancers; and the tumour necrosis factor (TNF) family, which have a wide range of effects on immune functioning, including in tissue modeling and remodelling, and neuronal development. They are often categorised as pro-inflammatory or anti-inflammatory[2], although as aforementioned most cytokines exhibit pleiotropy and therefore may have either effects depending on factors such as the site of expression and/or cell target. Blood-borne cytokines are able to pass the blood-brain barrier to enter cerebrospinal fluid and interstitial fluid spaces of the brain[3], although they are also produced in the brain by various cells such as neurons, astrocytes and microglia. Cytokines have been implicated in eating disorders due to their effects on appetite, body weight and eating behaviours, which may be mediated by their effects on brain areas such as the hypothalamus[4].

Three meta-analyses have quantified differences in concentrations of cytokines between ED populations and healthy controls[5–7], one of which examined only AN samples[5]. When comparing AN to healthy controls, these studies have found small-to-moderate sized increases in levels of the pro-inflammatory cytokines IL-1ß, IL-6 and TNF-α[5,6] and TNF-receptor-II[5] in AN. In a meta-analysis of inflammatory markers in EDs as a broader group, it has also been found that levels of TNF-α, and IL-1β are elevated in comparison with healthy control individuals (HCs)[7]. This has led to propositions that increases in pro-inflammatory cytokines may be a factor implicated in the aetiology of AN[8] and neuroinflammation is often cited as a possible aetiological factor in EDs (e.g[9]). However, this is not without debate, as several recent studies in both adolescents and adults with AN have failed to fully replicate these results and sources of this heterogeneity (moderators) are thus far unclear[10–14].

This heterogeneity may be explained by differences in sample characteristics (e.g., age, illness duration) and study factors (e.g., controlling or not controlling for factors such as age, tobacco use, use of pharmacological medications)[6]. Most of the studies included in the 2015 and 2018 meta-analyses[5,6] did not control for confounding clinical and lifestyle factors that may affect cytokine concentrations, such as age, menstruation, smoking status, medication usage, recent food intake, exercise, body fat, recent illness and concurrent medical (particularly inflammatory) or mental health diagnoses[15–20]. Since our last meta-analysis in 2018[6], several papers have been published that control for, or consider, such variables, which have had contrasting findings. Therefore, it is conceivable that the inclusion of newer studies controlling for such influences, or examining the effect of study quality, may modulate the overall effect sizes.

A prior meta-analysis found no changes after weight gain (body mass index [BMI] $\geq 17.5$ kg/m$^2$) in IL-6, TNF-α or IL-1β in people with AN followed longitudinally across three studies[5]. However, several studies with longitudinal designs over the course of weight restoration have since been published, enabling analyses with improved power. In conditions such as AN, it is possible that concentrations of cytokines are modified as dietary intake is reinstated[8], and it has also been suggested that reductions in neuroinflammation may contribute to weight gain[9].

This study aims to build upon a prior meta-analysis published by our group[6], asking the following additional research questions: (i) Including new evidence, do cytokine concentrations differ between people diagnosed with an ED and healthy individuals?; (ii) Using machine learning approaches, are there any moderators that explain the observed heterogeneity between studies?; (iii) How do between-group effect sizes vary as a function of study quality or publication date?; (iv) In ED participants, do cytokine concentrations change as a function of weight gain and/or symptom improvement? Here we show that when including new studies, and controlling for outliers, a small-sized increase in IL-6 is found in acute AN which may decrease longitudinally. However, the findings for increases in other pro-inflammatory cytokines such as TNF-α and IL-1β, in AN, are less robust. Additionally, we show that across studies, concentrations of IL-15 are increased and IL-7 decreased in people with AN compared to controls.

## Methods

The reporting of this systematic review was guided by the standards of the Preferred Reporting Items for Systematic Review and Meta-Analysis (PRISMA) Statement[21]. The quality assessment of the studies was conducted according to a modified version of the Newcastle-Ottowa Scale (NOS[22]). The full quality appraisal of the studies can be found in the Supplementary Information and Supplemental Data 1.

The study protocol was prospectively registered on the Open Science Framework, which can be found at https://osf.io/g6d3f.

### Literature search

Two reviewers (JLK and CB) independently searched four electronic databases (PubMed, Web of Science Core Collection, MEDLINE and PsycINFO) from the date of the last review, 4th May 2018, until 10th November 2024. An identical search strategy from the previous review by Dalton et al.[6] was replicated for this review. In short, searches included the following keywords, mapped to Medical Subject Heading with the Explode function where possible: eating disorder*, anorexia nervosa, bulimi* or binge eat* in combination with cytokine* chemokine*, inflammat*, interleukin, interferon, IFN, tumour necrosis factor, TNF, transforming growth factor or TGF. Searches were supplemented using the ascendancy approach (hand-searches of reference lists of relevant papers and reviews) and the descendancy approach (citation tracking in Google Scholar, Crossref).

### Eligibility criteria

Studies in any language of any study design that assessed cytokine concentrations in the serum, plasma or cerebrospinal fluid (CSF) of individuals with a Diagnostic and Statistical Manual of Mental Disorders (DSM)[23,24] or International Statistical Classification of Diseases (ICD)[25,26] diagnosis of ED were eligible for inclusion. Studies were included if they reported cross-sectional comparisons of cytokine concentrations between an ED group and HCs, or longitudinal assessments at a minimum of two time-points, with BMI or ED symptoms assessed at both time points.

Studies were excluded if: (i) they did not report group comparisons or longitudinal measurements of cytokine concentrations; (ii) participants had an organic cause for their disordered eating e.g., cancer, immunological conditions, genetic disorder, etc.; (iii) the sample was comprised of animals; iv) they measured cytokine production or genetic expression but did not assess cytokine concentrations; or v) there was significant reported or suspected study overlap between publications, defined as the same patient and or control group. Review articles, meta-analyses, conference proceedings/abstracts, book chapters, and unpublished theses were also not included.

### Source selection

Titles and abstracts of publications were imported into Endnote, where duplicates were removed. During an initial screening stage, titles and abstracts were assessed against basic eligibility criteria and those deemed likely to be irrelevant (e.g., studies in animal samples, studies of other medical or psychiatric conditions, studies with ineligible study designs such as reviews) were discarded. Full-texts of the remaining articles were then assessed against the full eligibility criteria, with the reasons for study exclusion documented (Fig. 1). The search process was conducted independently by two reviewers (JLK and CB) and discrepancies in the eligibility

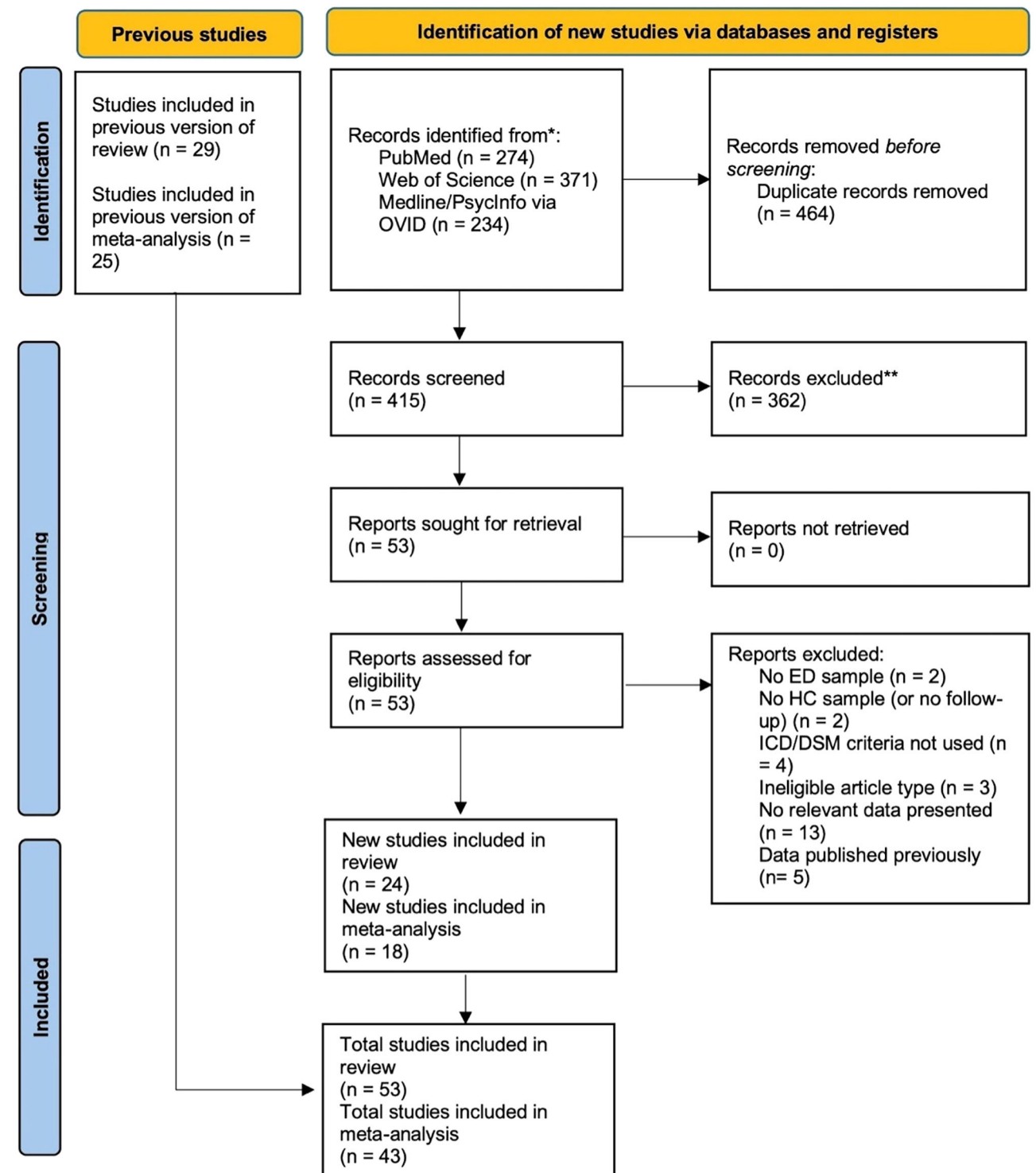

**Fig. 1 | PRISMA flow diagram.** ED eating disorder, HC healthy control, ICD International Classification of Diseases, DSM Diagnostic and Statistical Manual of Mental Disorders

assessment were discussed and resolved through discussion including a third reviewer (BD). The search results were then added to those obtained from the previous review by Dalton et al.[6] (i.e., studies published before 4th May 2018).

**Data extraction**

Two reviewers (JLK and CB) conducted the data extraction process and the extracted data was checked by one reviewer (JLK). At this stage, the references of the eligible publications were uploaded into an electronic summary table used in the previous study[6] and databases were combined for extraction. Extracted data included publication identifiers (study title, year, author list); sample characteristics, including sample size, demographics (e.g. mean ± standard deviation [SD] age), diagnostic criteria and clinical characteristics (e.g. mean ± SD illness duration, mean ± SD BMI), and medication status; parameters of interest, measurement methods, and concentrations of cytokines (mean ± SD), time interval between measurements (longitudinal studies, in weeks or days), content of intervention between measurements (longitudinal studies); potential confounds and

## Table 1 | Pre-specified moderators coded for meta-analysis

| Moderator | Codes |
|---|---|
| Region of study origin | Asia |
| | Europe |
| | North America |
| Year | - |
| NOS total score | - |
| Age group | Adolescents (12−18 years) |
| | Adults (≥18 years) |
| | Mixed |
| Sample fasting status | Fasted |
| | Not fasted/not reported |
| Type of blood sample | Serum |
| | Plasma |
| Use of psychotropic medication in the ED sample (%) | - |
| Percentage of smokers in sample (%) | - |
| Mean age of ED group (years) | - |
| Mean age of HC group (years) | - |
| Mean BMI of ED group (kg/m²) | - |
| Mean BMI of HC group (kg/m²) | - |
| Mean duration of illness in ED group (years) | - |

*BMI* body mass index, *ED* eating disorder, *HC* healthy controls, *NOS* Newcastle-Ottowa Scale.

control for confounds including presence/absence of smoking as an inclusion criteria, number of participants taking psychotropic medications (and types); main and additional study outcomes. Authors of potentially eligible publications were contacted where data were not reported ($n = 13$). In total, eight of these authors replied and provided data[11,13,27–32].

For studies that reported AN subtype data separately, the means and standard deviations were pooled for the AN meta-analyses. Standard errors (SE) were converted to SDs using the formula $SD = \sqrt{N \times SE}$, where N represents the sample size.

### Synthesis of results

Where studies were clinically homogeneous they were considered for pooling in a meta-analysis. Meta-analyses were performed for each cytokine where four or more studies were available for each ED separately. Where there were fewer than four studies for a given cytokine or raw data could not be obtained from authors, results were narratively synthesised.

### Statistics and reproducibility

Individual meta-analyses for each cytokine were conducted in Stata 18.0[33] using the 'metan' command. The main outcome measure was cytokine concentrations (pg/ml or ng/ml). Random effects-models using the Der-Simonian & Laird method[34] were used for all meta-analyses to pool the standardised mean difference (SMD) of the studies relative to the study sample size. The SMD of each study reflects the size of effect between the comparator groups in each study (i.e., ED vs. HC, or ED baseline vs. follow-up) relative to the variability observed in the study. All longitudinal meta-analyses utilised the two available time-points, or where multiple time-points were available, utilised the follow-up time-point representing weight restoration or discharge from care. Statistical significance of the overall SMD was ascertained according to the $p < 0.05$ threshold.

Outliers were identified when the confidence intervals (CI) of the individual study SMD did not overlap with the CI of the pooled effect (i.e., where the upper CI bound of the individual study effect is smaller than the lower CI bound of the pooled effect, or the lower CI bound of the individual study effect is bigger than the upper CI bound of the pooled effect).

Sensitivity analyses removing identified outliers were conducted and SMDs were re-estimated.

Publication bias was explored using the Egger's test for small study effects and funnel plots (Figs. S1–S16). The Duval and Tweedie trim and fill method[35] was used to identify smaller studies causing funnel plot asymmetry, adjust for this asymmetry, and re-estimate SMDs after adjusting for missing studies.

**Cumulative meta-analysis.** To explore how the effect sizes evolved over time for the cross-sectional comparisons of the main pro-inflammatory cytokines previously synthesised (IL-6, TNF-α and IL-1β) between AN and HC, cumulative meta-analyses were performed using NOS study quality total score, and year of publication as ordering variables. This enables an examination of how the effect size changes when adding new studies, starting from a) the newest studies to the oldest studies, and b) the highest quality studies to the lowest quality studies. Cumulative meta-analyses were displayed as forest plots (Figs. S17–20) and were interpreted by visual inspection.

**Between-study heterogeneity.** Between-study heterogeneity was assessed with the Higgins $I^2$ statistic, based on the Cochrane's Q (chi-square) statistic, which represents the proportion of total variation across studies that is attributable to heterogeneity rather than chance. Heterogeneity was considered to be substantial if above 50%, and considerable if above 75%. Tau$^2$, which estimates the amount of true variability between the SMDs of the included studies, was also reported.

Subgroup analyses were conducted with AN subtypes, where the study reported data for each subtype (i.e., AN-restricting type [-R] and AN-binge-eating/purging type [-BP]) individually. To further explore sources of heterogeneity, a random-effects MetaForest analysis was conducted in R version 4.4.2 (R Core Team, 2021). MetaForest is a machine-learning based exploratory approach to meta-analysis that mitigates overfitting and assesses the relative importance of potential moderators[36]. First, the approach ranks moderators in terms of their importance in predicting the effect size; and recursive preselection plots are generated. Using 100-fold replicated feature selection, moderators that reduced predictive performance (i.e., those with positive variable importance in fewer than 50% of replications) were excluded. The remaining moderators were then used to optimize the model (model tuning), consisting of 10,000 regression trees with random-effects weights, again with 100 replications to ensure robustness. The model's predictive performance was evaluated using the "out-of-bag" measure ($R^2_{oob}$), which estimates the variance explained by the moderators in a new dataset. A negative $R^2_{oob}$ value indicates that the model's predictions do not improve over simply using the average effect size as a predictor, suggesting that the moderators do not have a significant association with the effect size. Positive values indicate that the moderators explain variance in new data, and the most important moderators were then entered into meta-regressions. Analyses were only performed where there were 10 or more studies available. Pre-specified theoretical and methodological moderators are listed in Table 1.

### Reporting summary

Further information on research design is available in the Nature Portfolio Reporting Summary linked to this article.

## Results

### Characteristics of included studies

An additional 18 studies not included in the previous meta-analysis[6] were identified, resulting in a total of 43 studies from 11 countries included in the meta-analysis[11–13,27–32,37–70]. Of these, 40 were included in the cross-sectional meta-analyses comparing cytokine concentrations between AN and HCs, seven in the cross-sectional meta-analyses comparing BN with HCs, and 11 in the meta-analyses examining cytokine concentrations longitudinally in AN. The study and sample characteristics for studies included in the meta-

analyses are included in Supplemental Data 2. Findings of studies not included in the meta-analyses, due to insufficient data reporting or insufficient number of studies for meta-analysis[10,12,13,27-30,39-41,45,66,68,71-74](including cross-sectional comparisons between BED and HCs and recovered AN and HCs), are summarized in Supplemental Data 4 and the Supplementary Information.

Cytokines included in the cross-sectional meta-analyses between AN and HC were TNF-α, IL-1β, IL-6, IL-4, IL-7, IL-8, IL-10, IL-15, IFN-γ, transforming growth factor (TGF)-β and monocyte chemoattractant protein (MCP). Cytokines included in the cross-sectional meta-analyses between BN and HC were IL-6 and TNF-α. Longitudinally in AN, cytokines available for meta-analysis included TNF-α, IL-1β, and IL-6. Other cytokines were also measured in eligible studies, however sufficient data were not available to perform meta-analyses. Across studies, 31 studies measured cytokine concentrations peripherally in serum, 12 in plasma, and none in CSF.

The mean and SD age of participants with EDs and HCs was 21.7 ± 8.1 ($n$ = 32 studies) and 21.4 ± 6.7 ($n$ = 27 studies) years, respectively. All studies included only female participants. The mean BMI of the AN participants was 15.7 ± 2.3 kg/m$^2$ ($n$ = 34 studies), of BN participants was 31.7 ± 10.4 kg/m$^2$ ($n$ = 4 studies; 22.0 ± 3.1 kg/m$^2$ when excluding one study with a specifically bariatric population[66]), and of HCs was 21.2 ± 3.0 kg/m$^2$ ($n$ = 29 studies). Diagnosis was based on the DSM-V ($n$ = 14), DSM-IV ($n$ = 28) or DSM-III ($n$ = 1). Mean illness duration for ED participants, reported in 16 studies, was 5.2 ± 7.2 years. Medication usage of people in the ED groups was reported in 32 studies, and in 20 studies participants were confirmed medication-free as part of assessment or eligibility procedures. In the remaining 12 studies, medications used included anti-depressants, antipsychotics, sedatives, neuroleptics and anxiolytics.

For the longitudinal studies in AN, the mean BMI at baseline was 15.6 ± 1.9 kg/m$^2$ ($n$ = 8 studies) and at follow-up was 18.4 ± 2.0 kg/m$^2$ ($n$ = 7 studies). The interval between baseline and follow-up assessments used in the meta-analyses ranged from 2−12 months. The intervention used across studies included inpatient weight restoration treatment ($n$ = 5), specialist ED treatment ($n$ = 1), unspecified weight gain treatment ($n$ = 1) and with cognitive behavioural therapy (CBT; $n$ = 1), CBT alone ($n$ = 1), and CBT with pharmacological therapy ($n$ = 1). In one study, treatment was unspecified.

### Quality assessment of included studies
The rating of studies depends on three characteristics: the selection of study groups, the comparability of groups and the ascertainment of the outcome. Of the included studies, 26 were good quality, 24 were fair quality and three were poor quality (see Supplementary Information for the full description of the quality assessment). A detailed overview of the variables controlled for in individual studies is provided in Supplemental Data 3.

### Cross-sectional meta-analysis results
The results of the cross-sectional meta-analyses comparing participants with AN to HC, and participants with BN to HC, are displayed in Table 2.

### Anorexia nervosa
**Tumour-necrosis factor-α (TNF-α)**. Across 30 studies with a total of 1636 participants (AN $n$ = 872; HC $n$ = 764), six found elevated levels of TNF-α and none reported reduced levels (Fig. 2). Across all studies there was a small difference in TNF-α concentrations between AN and HCs whereby AN had elevated levels (SMD = 0.27; 95% CI 0.01, 0.54; $p$ = 0.040). Heterogeneity estimates were high ($I^2$ = 84%; Table 2) and although there was no evidence of publication bias from the Egger's test, there was clear funnel plot asymmetry with several studies showing an abnormally high SMD (Fig. S1). Subsequently six outliers were identified[12,32,40,57,60,69]. After removing these outliers, the effect size became marginal and non-significant (SMD = 0.06; 95% CI –0.10, 0.22; $z$ = 0.71; $p$ = 0.475) and the $I^2$ heterogeneity estimate reduced to 47.26%.

### Sensitivity and moderator analyses
When removing studies where >10% of the sample was using psychotropic medication[11,12,38,47,59,64], the SMD increased to 0.41 ($p$ = 0.011). Heterogeneity tests remained consistent ($I^2$ = 85%; Egger's test $z$ = 0.99, $p$ = 0.322).

A total of 15 studies reported AN subtype data for TNF-α, which indicated no subgroup difference between the AN-R (AN-R $n$ = 264; HC $n$ = 315) and AN-BP (AN-BP $n$ = 40; HC $n$ = 182) meta-analyses ($p$ = 0.690; Fig. S21). When including only these studies, neither meta-analyses showed significant differences between the individual subgroups (i.e., AN-R or AN-BP) and HCs.

Cumulative meta-analyses were conducted using year and NOS total study quality score as sorting variables in a descending fashion. A visual inspection highlighted that newer studies were associated with lower SMDs between AN and HC for TNF-α (Fig. S17), although the pattern for the NOS score was less clear (Fig. S18).

To further investigate sources of heterogeneity, an exploratory random-effects MetaForest analysis was conducted after removing study outliers. The analysis identified eight potential moderators (age of the AN sample, age of the HC sample, percentage of the sample using medication, age group, percentage of the sample who were smokers, sample type (serum/plasma), study quality score and year of publication; plots shown in Figs. S22–24), however none showed significant associations when entered into meta-regressions (Table S1). The full details can be found in the Supplementary Information.

**Interleukin-6 (IL-6)**. Across 25 studies including a total of 1596 participants (AN $n$ = 822, HC $n$ = 774), nine exhibited elevated levels and two exhibited reduced levels of IL-6 in AN (Fig. 3). The pooled mean concentrations of IL-6 were significantly higher in AN compared to HC with a small effect (SMD = 0.36; 95% CI 0.11, 0.61; $p$ = 0.005; Table 2; Fig. 2). This analysis showed high heterogeneity ($I^2$ = 82%) and the Egger's test for small study effects was significant. Five studies were identified as outliers following an inspection of the 95% CIs[11,12,32,40,60] and the funnel plot (Fig. S2). When removing these studies the SMD was slightly lower but remained significant (SMD = 0.32; $p$ = 0.001); the $I^2$ statistic lowered to 65.1% and the Egger's test was no longer significant ($z$ = 1.98, $p$ = 0.050).

### Sensitivity and moderator analyses
When removing studies where >10% of the sample were using psychotropic medication[11,12,38,47,59,64], the SMD increased to 0.50 and remained significant ($p$ < 0.001), although heterogeneity remained high ($I^2$ = 82%) and the Egger's test was significant ($z$ = 2.33; $p$ = 0.020).

A total of 15 studies reported AN subtype data, which indicated no subgroup difference between the AN-R (AN-R $n$ = 370; HC $n$ = 433) and AN-BP (AN-BP $n$ = 36; HC $n$ = 180) meta-analyses ($p$ = 0.790; Fig. S25). With this fewer number of included studies, neither meta-analysis showed significant differences between the individual subgroups (i.e., AN-R or AN-BP) and HCs.

In cumulative meta-analyses, both forest plots indicated that higher quality studies and newer studies had overall lower SMDs between AN and HC (Figs. S19 and S20).

After removing study outliers, an exploratory random effects MetaForest analysis identified the following moderators as the most important in explaining the effect size: study year, NOS score, study region, age group, percentage of sample using medication, fasting status, mean age, BMI, and illness duration of the AN sample, and mean age and BMI of the HC sample (Supplementary Information; plots shown in Figs. S26–28). These moderators were entered into meta-regressions (Table S2). Only study region was significant, with studies conducted in North America showing a greater difference between AN and HC compared to studies in Europe (B = 0.78; SE = 0.29; $z$ = 2.72; $p$ = 0.007; 95% CI 0.22, 1.34).

**Interleukin-1β (IL-1β)**. Across 14 studies with a collective sample size of 598 participants (AN $n$ = 354, HC $n$ = 244), there was a non-significant moderate-sized increase in pooled concentrations of IL-1β between AN and HC groups (SMD = 0.53, 95% CI -0.06, 1.12; $p$ = 0.080; $I^2$ = 91%;

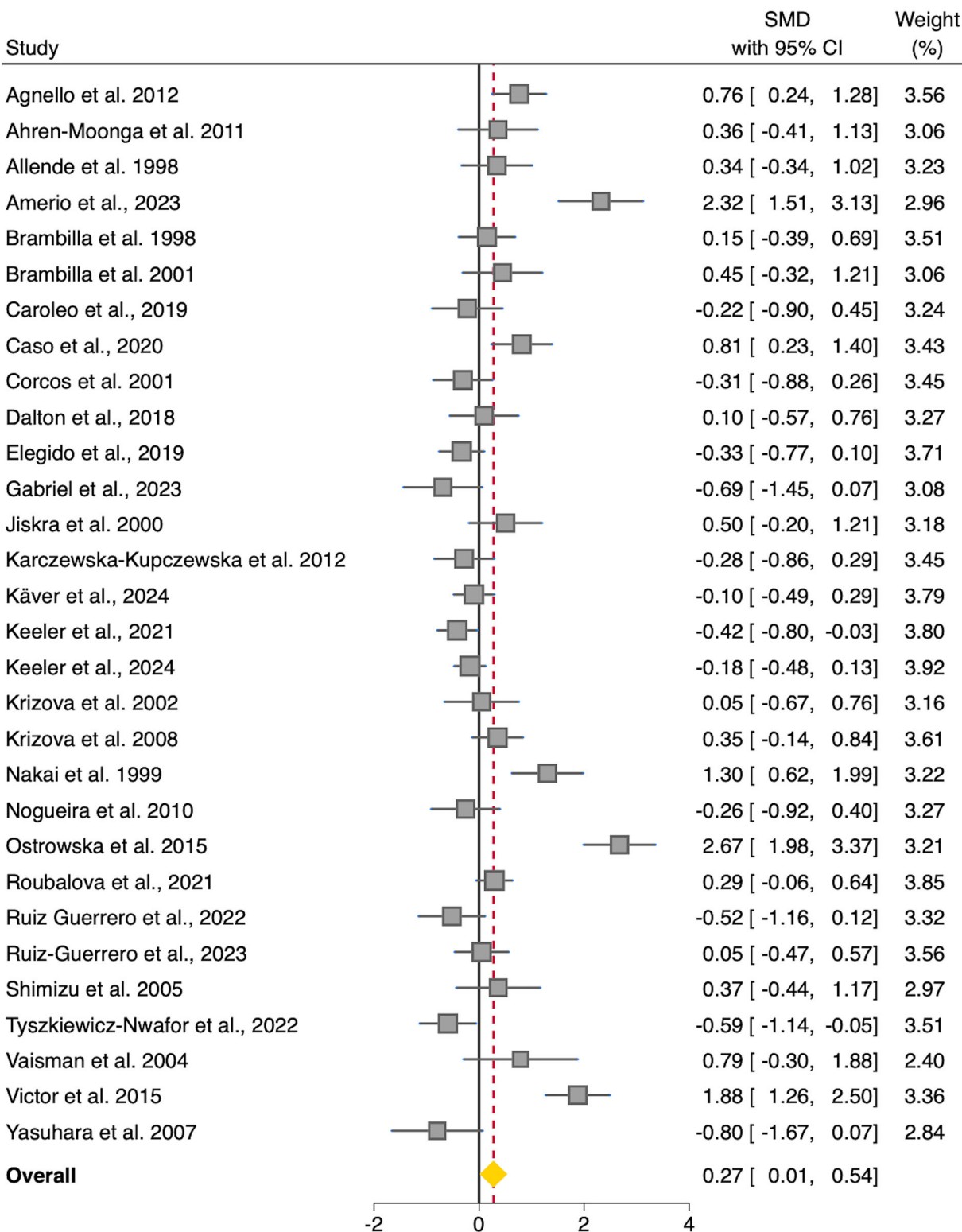

**Fig. 2 | Forest plot of standardized mean difference in tumour necrosis factor (TNF)-α between anorexia nervosa (AN) participants and healthy controls (HC).** $N = 30$ studies were included (AN $n = 872$, HC $n = 764$). Zero (black line) is the line of no effect, and studies to the right of zero indicate an elevation of TNF-α in AN compared to HCs. The red dotted line indicates the overall effect size. When removing outliers[12,32,40,60,69] the SMD reduced to 0.06 and became non-significant ($p = 0.475$). Error bars reflect 95% CIs. SMD = standardized mean difference; CI = confidence intervals.

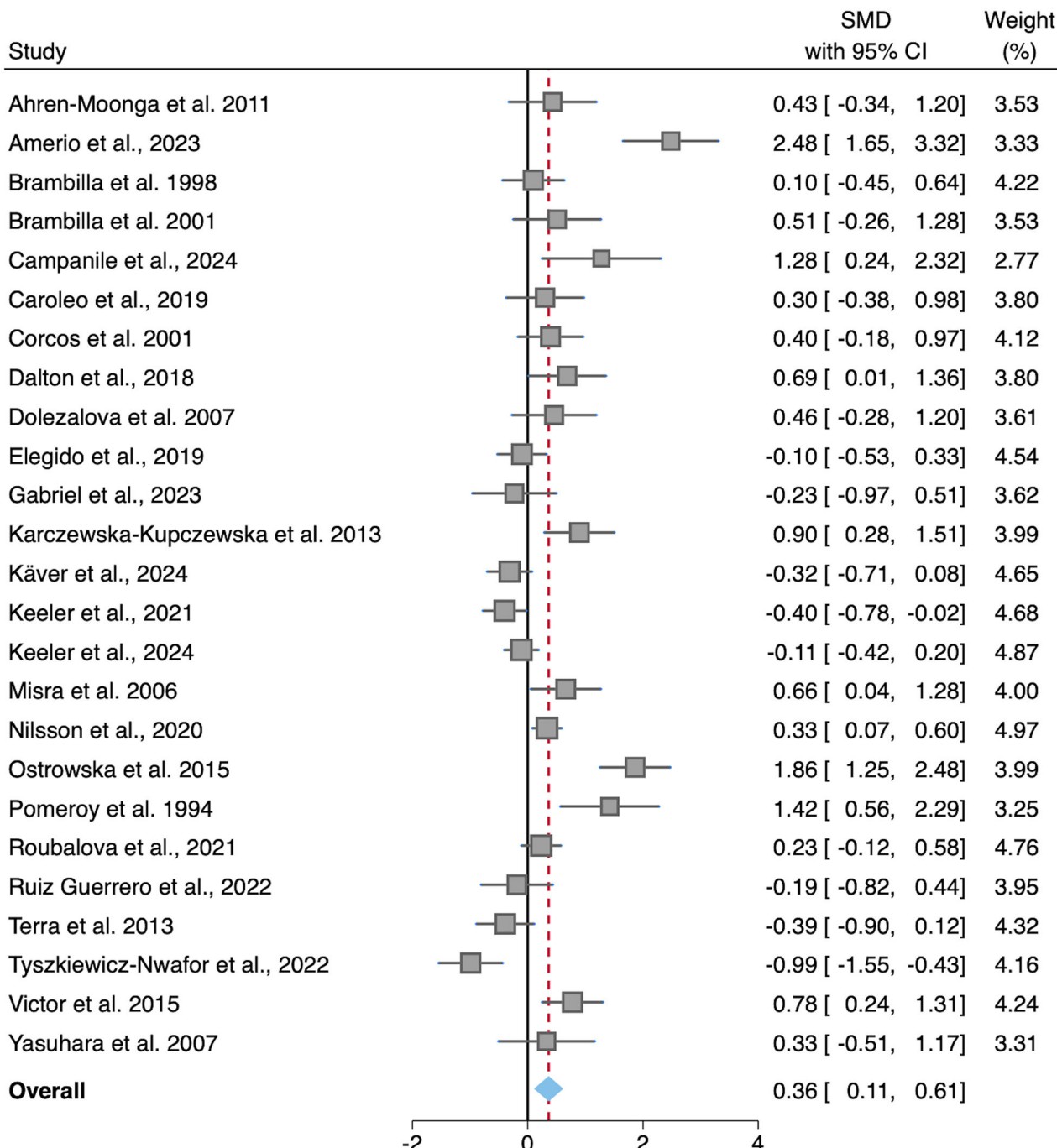

**Fig. 3 | Forest plot of standardized mean difference in IL-6 between AN participants and HCs.** N = 25 studies were included (AN n = 822, HC n = 774). Zero (black line) is the line of no effect, and points to the right of zero indicate an elevation of IL-6 in AN compared to HCs. The red dotted line indicates the overall effect size. When removing outliers[11,12,32,40,60] the SMD reduced to 0.32 but remained significant (p = 0.001). Error bars reflect 95% CIs. SMD standardized mean difference, CI confidence intervals.

Table 2; Fig. 4). From exploring the heterogeneity (95% CIs) and the funnel plot whereby two studies had abnormally high SMDs and one low (Fig. S3), three studies were identified as outliers[40,60,64]. The effect size was smaller and remained non-significant after removing these studies (SMD = 0.23; 95% CI –0.04, 0.50; z = 1.67; p = 0.095; $I^2$ = 44%).

**Sensitivity and moderator analyses**
There were three studies where >10% of the sample were using psychotropic medication[11,59,64]. When removing these studies, the SMD increased

substantially to 0.79 and became significant (p = 0.026), although heterogeneity estimates were similar ($I^2$ = 90%).

Subgroup analyses according to AN subtype were not possible, as few studies reported data per subtype. An exploratory random effects Meta-Forest analysis provided no evidence for associations between the entered moderators and the SMD for IL-1β (Supplementary Information; Fig. S29).

**Other cytokines available for meta-analysis.** Eight additional cytokines had at least four studies available and thus were meta-analysed (IL-4, IL-7,

**Fig. 4 | Forest plot of standardized mean difference in IL-1β between AN participants and HCs.** $N = 14$ studies were included (AN $n = 354$, HC $n = 224$). Zero (black line) is the line of no effect, and points to the right of zero indicate an elevation of IL-1β in AN compared to HCs. The red dotted line indicates the overall effect size. When removing outliers[40,60,64] the SMD reduced to 0.23 and remained non-significant ($p = 0.095$). Error bars reflect 95% CIs. SMD standardized mean difference, CI confidence intervals.

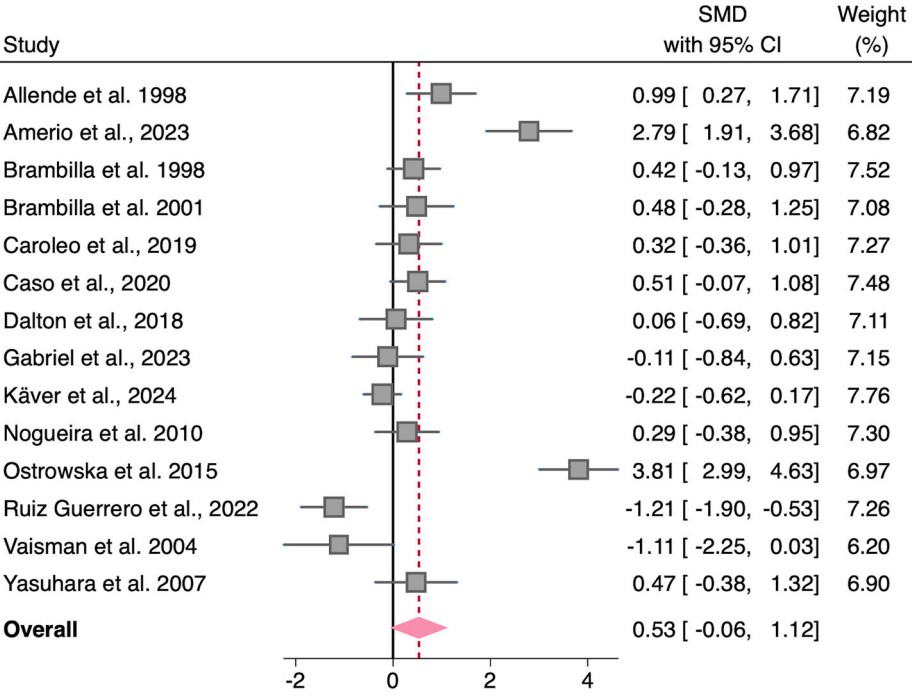

Random-effects DerSimonian–Laird model

| Study | SMD with 95% CI | Weight (%) |
|---|---|---|
| Allende et al. 1998 | 0.99 [ 0.27, 1.71] | 7.19 |
| Amerio et al., 2023 | 2.79 [ 1.91, 3.68] | 6.82 |
| Brambilla et al. 1998 | 0.42 [ -0.13, 0.97] | 7.52 |
| Brambilla et al. 2001 | 0.48 [ -0.28, 1.25] | 7.08 |
| Caroleo et al., 2019 | 0.32 [ -0.36, 1.01] | 7.27 |
| Caso et al., 2020 | 0.51 [ -0.07, 1.08] | 7.48 |
| Dalton et al., 2018 | 0.06 [ -0.69, 0.82] | 7.11 |
| Gabriel et al., 2023 | -0.11 [ -0.84, 0.63] | 7.15 |
| Käver et al., 2024 | -0.22 [ -0.62, 0.17] | 7.76 |
| Nogueira et al. 2010 | 0.29 [ -0.38, 0.95] | 7.30 |
| Ostrowska et al. 2015 | 3.81 [ 2.99, 4.63] | 6.97 |
| Ruiz Guerrero et al., 2022 | -1.21 [ -1.90, -0.53] | 7.26 |
| Vaisman et al. 2004 | -1.11 [ -2.25, 0.03] | 6.20 |
| Yasuhara et al. 2007 | 0.47 [ -0.38, 1.32] | 6.90 |
| **Overall** | 0.53 [ -0.06, 1.12] | |

IL-8, IL-10, IL-15, IFN-γ, MCP and TGF-β; Fig. S30 and S31, Table 2). Only concentrations of IL-15 and IL-7 were significantly different between AN and HC, whereby IL-15 was higher in the AN group with a moderate effect size (SMD = 0.67; 95% CI 0.07, 1.26; $p = 0.029$; $I^2 = 83\%$) and IL-7 was lower in AN with a moderate effect size (SMD = -0.58, 95% CI -1.07, -0.09; $p = 0.020$; $I^2 = 76\%$). Concentrations of IL-4, IL-8, IL-10, IFN-γ, MCP and TGF-β were not significantly different between AN and HCs.

One study was identified as an outlier in the meta-analyses of IL-10 and IFN-γ[27], and the removal of this study reduced the SMDs to –0.04 (95% CI –0.28, 0.20; $p = 0.761$; $I^2 = 32\%$) and –0.11 (95% CI –0.51, 0.28; $p = 0.571$; $I^2 = 59\%$), respectively. One outlier was identified in the IL-8 meta-analysis[32], and the removal of this study increased the SMD to 0.13 (95% CI –0.40, 0.67; $p = 0.626$). Two studies[40,60] were identified as outliers in the TGF-β meta-analysis, and the removal of these lowered the SMD to –0.14 (95% CI –0.11, 0.84; $p = 0.783$).

According to $I^2$, estimated heterogeneity was high in the meta-analyses of IL-7, IL-8, IL-10, IL-15, IFN-γ and TGF-β. The trim and fill method resulted in two imputed studies in both the IL-4 and MCP analyses (Figs. S32 and S33); adjusting for these imputed studies altered the SMDs to –0.09 and –0.10, respectively (both still non-significant). Funnel plots for meta-analyses are included in the Supplementary Information (Figs. S4–11).

**Bulimia nervosa**
**Interleukin-6 (IL-6).** The meta-analyses of the studies ($n = 5$) in BN populations indicated higher concentrations of IL-6 in participants with BN ($n = 163$) compared with HCs ($n = 115$), although this was not significant (Table 2; Fig. 5). One study[66] was deemed an outlier, which is likely due to the higher mean BMI of the sample (41.1 kg/m²). Removing this study from the IL-6 meta-analysis reduced the pooled SMD to 0.30 (95% CI -0.25, 0.86; $p = 0.287$; $I^2 = 65\%$).

**Tumour necrosis factor-α (TNF-α).** There were significantly higher concentrations of TNF-α in participants with BN compared with HCs (Table 2; Fig. 5) across five studies (BN $n = 145$, HC $n = 123$). The heterogeneity estimates were considerably high for both analyses. The funnel plot indicated asymmetry (Fig. S13) which was confirmed by a significant

Egger's test for small study effects (z = 4.91, $p < 0.001$). The same study[66] was deemed an outlier and removing this study reduced the overall effect size (SMD = 0.72) rendering the difference between groups non-significant (95% CI -0.14, 1.58; $p = 0.101$) and slightly reducing the $I^2$ heterogeneity statistic to 84%. After removing this study the Egger's test was no longer significant (z = 1.30, $p = 0.192$).

**Longitudinal meta-analyses in AN**
There were sufficient data to meta-analyse differences between baseline and follow-up in individuals with AN undergoing weight restoration for the cytokines IL-6, IL-1β and TNF-α. All meta-analyses utilised the two available time-points, or where multiple time-points were available, utilised the follow-up time-point representing weight restoration or discharge from care. Insufficient studies were available to investigate longer-term follow-ups. Forest plots for the analyses can be seen in Figs. S34–36 and full results of the analyses can be seen in Table 3.

As sensitivity analyses, studies were removed from analyses where either ≤10% weight gain occurred, where participants didn't reach at least 80% ideal body weight, or where the follow-up group mean BMI was ≤17 kg/m², depending on what data was reported in the study. There were insufficient studies to explore moderators or conduct subgroup analyses.

**Tumour necrosis factor-α (TNF-α)**
The meta-analysis of TNF-α included 267 AN participants at baseline and 240 at follow-up (mean BMI increase 2.85 kg/m², reported by eight studies) from nine studies, finding no significant difference between time-points (SMD = -0.05, 95% CI -0.22, 0.13; $p = 0.617$ $I^2 = 0\%$; Table 3; Fig. S34). The follow-up period across studies ranged from 7.3−32.2 weeks (M ± SD = 14.1 ± 8.3 weeks). The trim and fill procedure imputed two missing studies (Fig. S37), increasing the SMD to –0.07 (95% CI –0.24, 0.10). The estimated SMD remained similar when removing three studies[28,65,70] where weight increase was insufficient (SMD = –0.04; 95% CI –0.23, 0.15; z = -0.43; $p = 0.670$).

**Interleukin-6 (IL-6)**
Across eight studies with 279 AN participants at baseline and 239 participants at follow-up collectively (mean BMI increase 2.87 kg/m², reported by

**Fig. 5 | Forest plot of standardized mean difference in IL-6 and TNF-α between BN participants and HCs.** Zero (black line) is the line of no effect, and points to the right of zero indicate an elevation of the cytokine in BN compared to HCs. When removing a study outlier[66], effect sizes reduced to 0.30 and 0.72, respectively; both analyses were then non-significant. Error bars reflect 95% CIs. CI confidence intervals, IL interleukin, SMD standardized mean difference, TNF-α tumour necrosis factor-alpha.

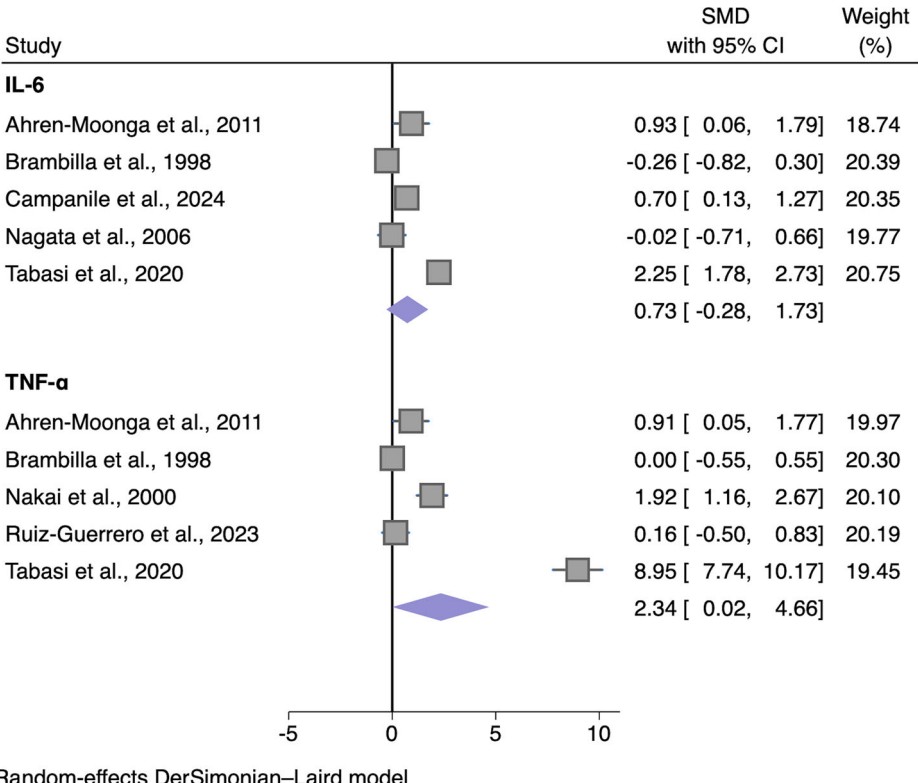

six studies), concentrations of IL-6 were significantly higher at baseline than follow-up with a small effect size (SMD = 0.21; 95% CI 0.01, 0.42; p = 0.042; $I^2$ = 22%; Table 3; Fig. S35). The follow-up period across studies ranged from 7.3−52 weeks (M ± SD = 16.8 ± 14.7 weeks). The Egger's test for small study effects was significant (Table 3) and the trim and fill procedure identified three missing studies (Fig. S38). When imputing these studies, the re-estimated SMD lowered to 0.10 and was non-significant (95% CI –0.13, 0.32). Additionally, when removing two studies where the BMI at follow-up was insufficient[28,70], the SMD reduced to 0.19 and again was non-significant (95% CI –0.05, 0.43; z = 1.52; p = 0.129).

### Interleukin-1β
Five studies with 125 AN participants at baseline and 104 participants at follow-up (mean BMI increase 2.40 kg/m², reported by four studies) were available for the meta-analysis of IL-1β concentrations. The follow-up period across studies ranged from 9.7 to 19.6 weeks (M ± SD = 13.3 ± 4.3 weeks). There was no significant difference between the baseline and follow-up time-points in IL-1β concentrations (SMD = 0.003; 95% CI -0.37, 0.37; p = 0.988; $I^2$ = 39%; Table 3; Fig. S36). After removing two studies where there was insufficient weight increase[28,70], the SMD increased to –0.10 but remained non-significant (95% CI –0.68, 0.48; z = -0.34; p = 0.733).

### Discussion
This updated systematic review examined cytokine concentrations in people with AN and people with BN compared with controls, and in AN across multiple time-points (largely, after weight restoration treatment). A total of 53 studies were included, 43 of which were entered into meta-analyses with a combined total of 2,533 participants. Cross-sectionally, in people with AN, concentrations of IL-6 and TNF-α were significantly higher with a small effect size, although differences in TNF-α became smaller and non-significant when removing study outliers. These analyses did not differ according to AN subtype, but when removing studies where > 10% of the sample were taking psychotropic medication, the effect size increased for IL-6, TNF-α and IL-1ß (the latter becoming

significant, with elevated levels in AN). Cumulative meta-analysis plots visually indicated that higher quality and more recently published studies produced smaller effect sizes. Concentrations of IL-15 were moderately higher in AN, and concentrations of IL-7 were moderately lower, compared with controls. Several other cytokines (IL-1β, IL-4, IL-8, IL-10, IFN-γ, MCP and TGF-β) were not significantly different between AN and controls. For the cross-sectional analyses of IL-6, TNF-α, and IL-1β, machine learning was used to explore sources of heterogeneity, although it did not identify any consistently important moderator. Longitudinally, meta-analyses were performed to examine differences in concentrations of IL-6, TNF-α and IL-1β from baseline to follow-up in AN, finding significant but small decreases over time only for IL-6, although this may be due to publication bias as this finding was no longer significant when adjusting for funnel plot asymmetry.

In BN, sufficient data were available for IL-6 and TNF-α, whereby TNF-α was significantly higher than controls with a large effect size, although critically this effect size substantially decreased and became non-significant when removing a study outlier.

The findings of this updated review broadly align with and extend the results of Dalton, et al.[6]. However, they also indicate the absence of a systemic inflammatory profile in AN, given that the majority of pro-inflammatory cytokines did not show elevations, and meta-analyses of other proteins that are indicative of inflammation, such as C-reactive protein, show decreases in AN[75]. Nevertheless, it is apparent that concentrations of IL-6 may be slightly elevated in the acute stages of AN and may decrease longitudinally. Interestingly, the pleiotropic functions of IL-6 include both pro-inflammatory and anti-inflammatory effects. Secreted by macrophages in response to pathogen-associated molecular patterns (PAMPs), it is known to mediate fever and the acute phase immune response, but also has inhibitory effects on TNF[76,77]. Relevant to AN, IL-6 has also been found to stimulate energy mobilization[78] and as a myokine is elevated up to one hundred times basal rate during exercise in response to muscle contraction[79]. Therefore, the factors contributing to increased IL-6 in acute AN could be manifold both as a result of the behavioural symptoms associated with AN and the adaptation of the body to starvation.

**Table 2 | Summary of comparative outcomes and heterogeneity for all conducted cross-sectional meta-analyses**

| Cytokine (n studies) | Sample N (ED, HC) | SMD | 95% CI | z | p | Heterogeneity | Egger's test |
|---|---|---|---|---|---|---|---|
| *Anorexia nervosa vs healthy controls* | | | | | | | |
| TNF-α (n = 30) | 872, 764 | 0.27 [a] | 0.01, 0.54 | 2.06 | 0.040 | Tau² = 0.43 I² = 84.05% | z = 1.63 p = 0.103 |
| IL-6 (n = 25) | 822, 774 | 0.36 [b] | 0.11, 0.61 | 2.82 | 0.005 | Tau² = 0.32 I² = 81.81% | z = 3.03 p = 0.002** |
| IL-1β (n = 14) | 354, 244 | 0.53 [c] | −0.06, 1.12 | 1.75 | 0.080 | Tau² = 1.14 I² = 90.59% | z = 0.57 p = 0.570 |
| IL-10 (n = 7) | 278, 250 | 0.41 [d] | -0.17, 0.99 | 1.39 | 0.166 | Tau² = 0.50 I² = 88.09% | z = 3.57 p < 0.001** |
| IFN-γ (n = 6) | 171, 146 | 0.37 [e] | −0.41, 1.15 | 0.92 | 0.356 | Tau² = 0.83 I² = 89.94 | z = 2.60 p = 0.009** |
| IL-8 (n = 5) | 235, 229 | -0.16 [f] | -0.78, 0.47 | -0.49 | 0.626 | Tau² = 0.44 I² = 88.83% | z = -0.21 p = 0.831 |
| MCP (n = 5) | 235, 229 | -0.19 | -0.44, 0.06 | -1.50 | 0.135 | Tau² = 0.03 I² = 33.07% | z = -2.29 p = 0.022* |
| TGF-ß (n = 5) | 240, 183 | 0.61 [g] | -0.90, 2.11 | 0.79 | 0.431 | Tau² = 2.75 I² = 96.76% | z = 3.39 p = 0.001** |
| IL-4 (n = 4) | 124, 105 | -0.01 | -0.28, 0.25 | -0.09 | 0.925 | Tau² = 0.00 I² = 0.00% | z = 1.29 p = 0.196 |
| IL-7 (n = 4) | 211, 188 | -0.58 | -1.07, -0.09 | -2.32 | 0.020 | Tau² = 0.18 I² = 75.55% | z = -0.55 p = 0.582 |
| IL-15 (n = 4) | 183, 126 | 0.67 | 0.07, 1.26 | 2.19 | 0.029* | Tau² = 0.30 I² = 82.74% | z = 1.96 p = 0.050 |
| *Bulimia nervosa vs healthy controls* | | | | | | | |
| IL-6 (n = 5) | 163, 115 | 0.73 [h] | -0.28, 1.73 | 1.42 | 0.157 | Tau² = 1.21 I² = 92.76% | z = -0.48 p = 0.632 |
| TNF-α (n = 5) | 145, 123 | 2.34 [i] | 0.02, 4.66 | 1.98 | 0.048* | Tau² = 6.84 I²: 97.88% | z = 4.91 p < 0.001** |

[a] When outliers removed SMD = 0.06, *p* = 0.475 (TNF-α AN); [b] When outliers removed SMD = 0.32 (IL-6 AN); [c] When outliers removed SMD = 0.23 (IL-1β AN); [d] When outlier removed SMD = −0.04 (IL-10 AN); [e] When outlier removed SMD = −0.11 (IFN-γ AN); [f] When outlier removed SMD = 0.13 (IL-8 AN); [g] When outliers removed SMD −0.15 (TGF-β AN); [h] When outlier removed SMD = 0.30 (IL-6 BN). [i] When outlier removed SMD = 0.06, *p* = 0.475 (IL-6 TNF-α).* Significant at the *p* < 0.05 threshold; ** Significant at the *p* < 0.001 threshold. *CI* confidence intervals, *ED* eating disorder, *HC* healthy control, *IFN-γ* interferon-gamma, *IL* interleukin, *MCP* monocyte chemoattractant protein, *p* p-value, *SMD* standardized mean difference, *TNF-α* tumour necrosis factor-alpha, z z-score.

**Table 3 | Summary of comparative outcomes and heterogeneity for longitudinal meta-analyses in AN samples**

| Cytokine (n studies) | Sample N (T0, T1) | SMD | 95% CI | z | p | Heterogeneity | Egger's test |
|---|---|---|---|---|---|---|---|
| TNF-α (n = 9) | 267, 240 | -0.05 [a] | -0.22, 0.13 | -0.50 | 0.617 | Tau² = 0.00 I² = 0.00% | z = 0.64 p = 0.524 |
| IL-6 (n = 8) | 279, 239 | 0.21 [c] | 0.01, 0.42 | 2.03 | 0.042* | Tau² = 0.02 I² = 22.06% | z = 2.66 p = 0.008** |
| IL-1β (n = 5) | 125, 104 | 0.003 [b] | -0.37, 0.37 | 0.01 | 0.988 | Tau² = 0.07 I² = 38.61% | z = -1.48 p = 0.139 |

[a] When studies with insufficient weight gain removed SMD = −0.04 (TNF-α); [b] When studies with insufficient weight gain removed SMD = -0.10 (IL-1β); [c] When studies with insufficient weight gain removed SMD = 0.19 (IL-6). * Significant at the *p* < 0.05 threshold; ** Significant at the *p* < 0.001 threshold. *CI* confidence intervals, *IL* interleukin, *p* p-value, *SMD* standardized mean difference, T0 baseline time-point, T1 follow-up time-point, *TNF-α* tumour necrosis factor-alpha, z z-score.

Across the longitudinal studies, the average BMI increase ranged from 2.40 kg/m² to 2.87 kg/m², although in approximately half of studies in each analysis, the follow-up BMI did not surpass 18.5 kg/m². Therefore, it is possible that changes from baseline to follow-up may have been more pronounced with more consistent weight restoration. With further longitudinal studies including multiple time-points, future meta-analyses could explore the temporal dynamics of cytokine concentrations according to different BMI thresholds. Additionally, there was considerable variability in the time interval between baseline and follow-up (7.3−52 weeks across all studies), which may have contributed to our findings. The current evidence may signal that there is an absence of a substantial change in cytokine levels longitudinally, although more research is required.

Additionally, concentrations of IL-15 and IL-7 showed moderately-sized increases and decreases, respectively, in the acute stages of AN. Furthermore, one study included in this meta-analysis in adolescents with AN

found elevated IL-15 at baseline, discharge (weight restoration) and after a 1 year follow-up[11]. IL-15 induces the proliferation of natural killer cells in the innate immune system[80] but has both pro- and anti-inflammatory effects depending on its expression and site of action. It has been suggested that increases in IL-15 have a pathophysiological role in AN, specifically pertaining to its anabolic role in maintaining muscle mass in the case of acute illness and starvation[11]. This adaptation hypothesis would however not explain the elevated concentrations seen after weight restoration and after long-term follow-up in the aforementioned study[11]. Although, it could be related to exercise levels as IL-15 is upregulated after acute and chronic exercise[81]. IL-15 also modulates serotonergic transmission[82] and synaptic GABAergic transmission in the hippocampus, impairing short- and long-term episodic memory[83], and thus may be involved in the mood and memory disturbances often seen in AN. IL-7 has essential roles in providing survival and growth signals for different immune cells, and low levels can

**Table 4 | Recommendations for standardised reporting of biological data in people with eating disorders**

| Methodological consideration | Description |
|---|---|
| Open-access data sharing | Publish anonymised participant-level data or group-level data open-access according to Open Science principles. |
| Reporting on data distribution | Report information (or plots) on the distribution of the data across samples. |
| Reporting age | Report the mean and standard deviation age of the samples and subgroups, including specifying the age range. |
| Reporting other variables | Report the mean and standard deviation BMI, body fat percentage, illness duration. Additional variables that ideally should be reported include: medication usage (and type), number of smokers in each sample, mean and standard deviation values of psychopathological measures (e.g., the EDE-Q for ED symptoms, DASS for depression and anxiety symptoms), alcohol use (or exclusion based on alcohol misuse). |
| Fasting status | Ideally ensure that blood samples are collected in a standardised manner, for example after an overnight fast. If fasting has not occurred, report this transparently. |
| Time of sampling | Report the time at which the sample was collected, and preferably collect the sample at consistent times across participants and time-points (e.g., in the morning following an overnight fast). |
| Immunological status | Consider removing participants who have had recent infection or who have a comorbid inflammatory/autoimmune disorder. If included, report data separately for these individuals in the supplementary materials. |
| AN subtype | Report data separately according to AN subtype (e.g., in the supplementary materials). |
| Use of psychotropic medication | Report data separately according to usage of psychotropic medication classes (e.g., SSRIs, antipsychotics). |
| Menstrual cycle | Report menstrual phase for female participants if possible, and/or levels of hormones such as estradiol and progesterone. |
| Exercise | Consider including self-report or objective measures of exercise. |

*AN* anorexia nervosa, *BMI* body mass index, *DASS* Depression, Anxiety and Stress Scale, *ED* eating disorder, *EDE-Q* Eating Disorder Examination-Questionnaire.

produce immune deficiencies due to lymphopenia[84], which is seen in AN and is thought to be a result of malnutrition[85,86]. Indeed, reduced levels of IL-7 are not observed in constitutional thinness where individuals are underweight without a change in eating behaviour[72], and in AN, a longitudinal study has found increases in IL-7 with weight gain[29].

It is notable that in the 7 years since the previous meta-analysis was published[28], to the best of our knowledge only two new studies published in BN samples were eligible for inclusion in the meta-analyses, despite calls for more such studies in this population and in people with BED. The results including these new studies were broadly aligned with the previous meta-analysis, when removing one study that constituted a clear outlier. Overall, the pro-inflammatory cytokines TNF-α and IL-6 were similar between people with BN and controls. Moreover, one additional study found a decrease in the anti-inflammatory cytokine IL-10 in BN compared to BMI-matched controls[66]. In BED, there were insufficient studies to perform meta-analyses. A single study found a decrease in IL-10 in people with BED, although this was also found in individuals with obesity who did not have BED[27]. In another study, TNF-α was found to be elevated in BED compared to non-BED controls with a similar BMI[71], although IL-6[71] and IL-1ß[27] were similar between BED and controls. Therefore, decreases in IL-10 in BN and increases in TNF-α in BED may be BMI-independent findings in these populations, although more research is necessary.

This study was a well-controlled and pre-registered systematic review and meta-analysis of cytokine concentrations including the most studies of any meta-analysis published thus far, to the authors' knowledge. Authors were contacted where data were unavailable in publications, enabling the inclusion of eight additional studies. We were able to expand on the previous meta-analysis both by (a) including new studies of cytokines that have previously been meta-analysed; (b) meta-analysing several new cytokines that have not been previously examined, to the best of our knowledge; (c) examining changes in pro-inflammatory cytokines longitudinally in people with AN; and (d) utilising various approaches, including machine learning, to examine potential sources of heterogeneity. For transparency, these moderator analyses employing machine learning were decided for after pre-registration.

There are several limitations associated with the study design and conduct of the study. Firstly, because of the nature of cytokine data, data distributions across samples often showed skewedness, and therefore authors publish median rather than mean averages for their samples. Where this occurred, authors were contacted for mean and standard deviation values for samples to enable inclusion in the meta-analysis.

One acknowledgement of this approach is that meta-analyses of cytokines including such studies may be inherently biased, although an advantage of including these studies is that the analyses have increased power. Likewise, removing study outliers may introduce bias into results as it is possible that extreme results may constitute real biological anomalies. Therefore, we opted to report results before and after removing outliers. Finally, the results of the cumulative meta-analyses were interpreted visually, therefore the conclusions drawn from these analyses should be taken with caution.

Other limitations relating to the included studies themselves pertain mostly to a lack of data reporting, as aforementioned. Not all studies, for example, reported the fasting status of their sample, or the age range of their sample, meaning that several moderators had missing data. Therefore, the predictive capability of the machine learning approach to moderator identification, and the follow-up meta-regressions, was limited. There was also a paucity of longitudinal studies reporting cytokine concentrations after long-term follow-ups or prospectively over the longer-term course of the illness, meaning that it was not possible to examine trajectories over illness stages (e.g., in the case of longstanding, protracted, or severe-enduring illness). Likewise, we could unfortunately not perform subgroup analyses by BMI category, sex or psychotropic medication status as data were not reported in the studies. It is likely that these are important moderators given that cytokine levels fluctuate according to weight[87], menstrual cycle stage and sex[88,89], and psychotropic medication usage[18].

## Conclusions and future directions

Overall, it can be concluded that increases in IL-6 in the acute stages of AN are more robust than increases in TNF-α, and increases in IL-15 and decreases in IL-7 are apparent in AN compared with controls. Concentrations of IL-6 decrease longitudinally in people with AN, which may be associated with weight recovery, although this may also be due to publication bias and thus needs further investigation. When considering the effects of outliers, other cytokines (IL-1β, IL-4, IL-8, IL-10, IFN-γ, MCP, TGF-β and TNF-α) were not altered in AN. There were no differences between people with BN and healthy controls in IL-6 and TNF-α. Heterogeneity was generally high across analyses, the source of which remains unclear, although there appears to be a trend whereby higher quality and more recent studies showed smaller differences.

Notably, not all studies reported key data that enabled more in-depth moderator or subgroup analyses examining key variables known to influence immune functioning, such as smoking status[19], use of psychotropic

medications[18], time of sampling (circadian rhythms)[90–92], and fasting status/ dietary habits[93]. The present study found an increase in differences between AN and HC in IL-6, TNF-α and IL-1ß when samples with a high usage of psychotropic medication were excluded, likely as the usage of medications such as selective serotonin reuptake inhibitors (SSRIs) is associated with lower levels of IL-6, TNF-α and IL-1ß[94]. Therefore it is particularly important that studies report data according to whether individuals are or are not taking psychotropic medication, or consider this as an exclusion criterion. With this in mind, in Table 4 we have listed further recommendations to enable standardised reporting across studies to enable future high-quality and well-powered meta-analyses, individual participant data meta-analyses, and detailed moderator analyses of cytokine and other biological data in ED populations.

We reiterate from our previous publication[6] that studies in ED diagnoses under-represented in this field of research, such as people with BN, BED, ARFID, and atypical AN, are needed. Studies should also carefully consider eligibility criteria for the research to minimise the confounding influence of comorbid illness (e.g., recent infection or inflammatory/auto-immune conditions) or lifestyle factors (e.g., smoking, alcohol misuse), as well as methodological factors, and transparently report key variables relating to their samples (Table 4).

As a final remark, alterations in the concentration of other immune molecules such as immunoglobulins and C-reactive protein have also been suggested to be associated with AN and other EDs. For example, immunoglobulin G (IgG) seems to play a role as α-melanocyte stimulating hormone (α-MSH)-binding protein in the blood, and its levels have been found to be low in patients with AN at hospital admission but to increase alongside weight recovery[95]. Alterations in IgG autoantibodies directed to ghrelin and leptin might also be involved in the development of EDs[96]. Additionally, immunoglobulin A (IgA) concentrations have been found elevated in the saliva of patients with AN[97]. As aforementioned, C-reactive protein has been found to be decreased in AN[75]. Thus, for a comprehensive understanding of changes in immune molecules within patients with EDs, the complex relationships between cytokines, IgG, IgA, C-reactive protein and other molecular and cellular immune markers should be considered. Principal component analyses or machine learning approaches might help identifying subtypes with shared immunological profiles. The identification of such subtypes in future research might be more meaningful for the development of individually tailored biological treatments for people with EDs than a comparison of mean cytokine levels.

## Data availability

The source data for Figs. 2, 3 and 4 are in Supplemental Data 5 (sheets 1, 2 and 3, respectively). The source data for Fig. 5 is in Supplemental Data 5, sheet 5. The source data for Figs. S30 and S31 is in Supplemental Data 5, sheet 4. The source data for Figs. S34, S35 and S36 is in Supplementary Data 5 (sheets 6, 7 and 8, respectively). All other data are available from the corresponding author on reasonable request.

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

## Acknowledgements

JLK acknowledges financial support from a Medical Research Council (MRC) funded Doctoral Training Partnership stipend for commencing this project (ref: MR/N 013700/1). This study represents independent research part funded by the NIHR Maudsley Biomedical Research Centre at South London and Maudsley NHS Foundation Trust and King's College London. The views expressed are those of the author(s) and not necessarily those of the MRC, NIHR or the Department of Health and Social Care.

## Author contributions

J.L.K. contributed to the conceptualization and management of the study, the database searches, the data extraction, management and curation, the formal analysis, the interpretation of results, the writing of the protocol and manuscript, and the manuscript submission; C.B. contributed to the database searches, the data extraction, management and curation, the interpretation of results, and the writing of the manuscript; H.H., J.T., B.C. and U.S. contributed to the review and editing of the final manuscript; B.D. contributed to the conceptualization and supervision of the study, the data curation, the interpretation of results and the review and editing of the final manuscript.

## Competing interests

The authors declare no competing interests.
