## [Transparent Peer Review file · Communications Medicine]

Cytokine concentrations in people with eating disorders: A comprehensive updated systematic review and meta-analysis

Corresponding Author: Dr Johanna Keeler

Version 0:

Reviewer comments:

Reviewer #1

(Remarks to the Author)

I thank the editors of Communication Medicine for inviting me to peer review the manuscript “A comprehensive updated cross-sectional and longitudinal meta-analysis of cytokines in eating disorders.” This assignment has allowed me to deepen my understanding of my primary area of interest, eating disorders. Despite their severe consequences and high mortality rates, eating disorders are still considered a niche specialty and are often overlooked by journals (Marzola et al., 2022). These challenges in research dissemination may have serious implications for clinical advancement (Marzola et al., 2022).

This study presents a comprehensive meta-analysis examining cytokine levels in individuals with eating disorders (EDs) compared to healthy controls, with particular attention to subtype differences and longitudinal changes across time points. A total of 43 studies were included. In anorexia nervosa (AN) compared to healthy controls (HCs), IL-6 and IL-15 levels were higher, while IL-7 levels were lower. No differences were found for TNF- α , IL-1 β , IL-4, IL-8, IL-10, IFN- γ , MCP, and TGF- β when controlling for outliers. Longitudinally, IL-6 levels were lower at follow-up compared to baseline in AN, while TNF- α and IL-1 β remained unchanged. No differences emerged based on AN subtype. Individuals with bulimia nervosa (BN) showed no differences in IL-6 and TNF- α compared to HCs. The available data were insufficient to conduct meta-analyses for binge-eating disorder (BED) or other EDs.

This is a well-conducted meta-analysis of a large number of studies, offering a robust overview of cytokine alterations in EDs. The statistical methods employed are both innovative and appropriate, enhancing the reliability and validity of the findings. The Introduction clearly outlines the need for this research and provides a comprehensive overview of immunology in EDs. The Discussion thoughtfully contextualizes the findings within the broader literature, offering a clear interpretation of their clinical and theoretical implications.

I suggest the following minor points for consideration, listed in order of appearance, as hints for improving the manuscript:

Abstract:

1. The full name of “IFN- γ ” should be provided when first mentioned.
2. The text states: “Conclusion: In acute AN, concentrations of IL-6 and IL-15 are elevated and IL-7 is decreased, with evidence for normalisation of IL-6 over the course of weight restoration.” The findings, as explained in the Results, show a reduction with a small effect size; however, there is no mention of “normalisation”. Accordance should be reached.
3. Keywords should be added at the end of the Abstract.

Materials and Methods:

4. Page 6: The adherence to the PRISMA checklist and the quality assessment should be mentioned in the Abstract, if word count allows.
5. Page 7, section “2.3. Source Selection”: The sentence “The search process was conducted independently by two reviewers (JLK and CB)” should be complemented with information on how discrepancies were resolved.

Results:

6. In Table 2, I recommend adding a column indicating whether the studies controlled for clinical and lifestyle confounders that may affect cytokine levels, specifying which variables were considered.
7. Page 20, end of section "3.1. Characteristics of included studies": "CBT" appears twice, which may be unintended.
8. In the sections "3.3.1.1. Tumour-necrosis factor- α (TNF- α)" and "3.3.1.2. Interleukin-6 (IL-6)", the number of participants included in the AN subtype analysis should be reported.
9. In the sections "3.3.1.4. Other cytokines available for meta-analysis" and "3.3.2. Bulimia nervosa", information regarding the number of studies and total participants should be included.
10. In the section "3.4. Longitudinal meta-analyses in AN", it is stated: "As sensitivity analyses, studies were removed from analyses where either $\leq 10\%$ weight gain occurred, where participants didn't reach at least 80% ideal body weight, or where the follow-up group mean BMI was $\leq 17\text{kg/m}^2$, depending on what data was reported in the study." Considering the global worsening of the clinical picture in cases of severe-enduring eating disorders (SE-ED) (Wonderlich et al., 2020), a complementary analysis focusing on time points without BMI improvement would have been valuable. If not feasible due to a lack of studies, this should be acknowledged as a limitation of the current literature, suggesting future research in this direction.
11. In the sections "3.4.1. Tumour necrosis factor- α (TNF- α)", "3.4.2. Interleukin-6 (IL-6)", and "3.4.3. Interleukin-1 β ", the time span between assessments should be reported. This information should also be addressed in the Discussion and Abstract. The same applies to the change in BMI between time points (ΔBMI), if available.

Discussion:

1. A further limitation of the existing literature – and a potential explanation for inconsistent findings – is that cytokine levels are influenced by circadian rhythms and may vary depending on the time of sampling (Nilsson et al., 2016; Zeng et al., 2024; Zielinski and Gibbons, 2022). I would suggest that the authors explicitly acknowledge this as a limitation, noting that the studies included did not control for time of blood collection. Additionally, I suggest recommending in Table 5 greater standardization on this point in future research.

References:

- Marzola, E., Panero, M., Longo, P., Martini, M., Fernández-Aranda, F., Kaye, W.H., Abbate-Daga, G., 2022. Research in eating disorders: the misunderstanding of supposing serious mental illnesses as a niche specialty. *Eating and Weight Disorders - Studies on Anorexia, Bulimia and Obesity* 27, 3005–3016. <https://doi.org/10.1007/s40519-022-01473-9>
- Nilsson, G., Lekander, M., Åkerstedt, T., Axelsson, J., Ingre, M., 2016. Diurnal Variation of Circulating Interleukin-6 in Humans: A Meta-Analysis. *PLoS One* 11, e0165799. <https://doi.org/10.1371/journal.pone.0165799>
- Wonderlich, S.A., Bulik, C.M., Schmidt, U., Steiger, H., Hoek, H.W., 2020. Severe and enduring anorexia nervosa: Update and observations about the current clinical reality. *International Journal of Eating Disorders* 53, 1303–1312. <https://doi.org/10.1002/eat.23283>
- Zeng, Y., Guo, Z., Wu, M., Chen, L., 2024. Circadian rhythm regulates the function of immune cells and participates in the development of tumors. *Cell Death Discov* 10, 199. <https://doi.org/10.1038/s41420-024-01960-1>
- Zielinski, M.R., Gibbons, A.J., 2022. Neuroinflammation, Sleep, and Circadian Rhythms. *Front Cell Infect Microbiol* 12. <https://doi.org/10.3389/fcimb.2022.853096>

Reviewer #2

(Remarks to the Author)

This systematic review and meta-analysis represents a valuable expansion of prior work on cytokine profiles in eating disorders, incorporating 24 additional studies (total N=43) to examine cytokine differences between anorexia nervosa (AN), bulimia nervosa (BN), and healthy controls, while tracking longitudinal changes during AN treatment. The key findings are:

- AN vs. HC: elevated IL-6 and IL-15, alongside reduced IL-7, with no differences in TNF- α or IL-1 β after removing outliers.
- Longitudinal AN: While IL-6 appeared to decrease post-treatment initially, this effect disappeared after trim-and-fill adjustment. TNF- α and IL-1 β showed no changes.
- BN vs. HC: No significant cytokine differences emerged after outlier adjustment.

There are some major issues, listed below:

- Results (3.4.2) and the discussion. Looking at Figure S24, I notice significant funnel plot asymmetry in your longitudinal IL-6 analysis. The trim-and-fill procedure clearly shows that imputed studies reduce the effect to non-significance, yet the main text doesn't adequately address this critical finding. I would advise revising your conclusions to acknowledge that the IL-6 "decrease with weight restoration" may be an artifact of publication bias. Consider discussing how selective reporting of positive longitudinal results could explain this pattern.
- Results (3.1). While I see you've included medication use in your meta-regressions (Table S2), the null result ($B=-0.01$, $p=0.202$) doesn't necessarily rule out confounding. For instance, Käver et al. (2024) reported 45% SSRI use, and SSRIs can independently modulate cytokines why I would encourage conducting sensitivity analyses excluding studies with $>20\%$ medicated participants. This would strengthen your conclusions.
- Results (3.3.1.2). Your MetaForest analysis (Figure S34) primarily identified regional differences (North America vs. Europe) rather than biologically meaningful moderators. While this is interesting, I think highlighting ML's utility may be misleading when the main finding reflects geographic heterogeneity in protocols rather than pathophysiology. I recommend reframing these results as exploratory and emphasizing that regional discrepancies likely reflect methodological differences rather than biological insights.

Minor issues:

- Table 1. I may have missed this but I am lacking a definition of adult and adolescent in terms of age ranges.
- Results (3.1, Table 2). Your BN sample's mean BMI of 31.7 ± 10.4 kg/m² seems unusually high and likely includes atypical cases. Looking at Figures S16–S17, I can see extreme outliers that probably reflect this heterogeneity. I would advise either excluding these outliers or stratifying your BN analyses by BMI (e.g., typical BN [BMI 18–25] vs. atypical [BMI ≥ 30]) to avoid conflating different phenotypes.
- Results (3.3.1.1); Supplementary Figures S5, S7. Your TNF- α and IL-1 β funnel plots show clear asymmetry suggesting publication bias, but this isn't mentioned in the main text. Thus I suggest to briefly describing these patterns in your Results section to enhance transparency.

Reviewer #3

(Remarks to the Author)

I read with much interest the article titled “A comprehensive updated cross-sectional and longitudinal meta-analysis of cytokines in eating disorders”. This study updates a prior meta-analysis by integrating 24 new studies examining cytokine levels in people with eating disorders, with a focus on anorexia nervosa. It finds that IL-6 and IL-15 are elevated, and IL-7 is reduced in acute anorexia, while most other cytokines do not show consistent changes. The analysis also includes longitudinal data showing that IL-6 levels tend to decrease during weight restoration, suggesting possible normalization of immune function over time. However, I noted some points that should be addressed by authors:

- 1- The inclusion of longitudinal data and application of machine learning (MetaForest) for moderator detection are substantial advancements. However, core findings on cytokines such as IL-6 and TNF- α remain similar to earlier reviews. Please emphasize the novel insights from longitudinal findings and machine-learning-based heterogeneity analyses more clearly in the abstract and discussion to highlight added value.
- 2- Strengthen the limitations section by explicitly stating the impact of female-only samples and variable control for confounders. Consider subgroup analysis where possible on medication status or BMI level. If there is no data just add to the limitations
- 3- Rephrase language suggesting causality or normalization unless directly supported by prospective trial data. Replace with terms such as "associated decrease" or "observed reduction over time."
- 4- The manuscript is clearly written with strong structure. However, the results section is densely packed with data and could overwhelm readers. It would be better to move some detailed statistical figures or forest plots into supplementary materials and condense result presentation into clearer thematic sub-sections.
- 5- Provide narrative justification when referencing prior meta-analyses or conflicting findings. Ensure figures use consistent formatting and define abbreviations within each figure/table legend.
- 6- Define each acronym at first mention in the abstract and main text. Add a glossary or abbreviation list if space permits.

Please provide a point-to-point response letter.

Version 1:

Reviewer comments:

Reviewer #1

(Remarks to the Author)

The authors have thoroughly addressed all the minor points raised in the initial review. The revised manuscript is now clear, coherent, and well-structured. It reflects the qualities expected of a high-quality scientific article. I have no further suggestions, and I believe the manuscript is ready for publication in its current form.

Reviewer #2

(Remarks to the Author)

The changes and updates made have improved the manuscript which now is good and acceptable with me. I have no further comments.

Reviewer #3

(Remarks to the Author)

I have no more points; all has been addressed by authors

Reviewer #1

I thank the editors of Communication Medicine for inviting me to peer review the manuscript “A comprehensive updated cross-sectional and longitudinal meta-analysis of cytokines in eating disorders.” This assignment has allowed me to deepen my understanding of my primary area of interest, eating disorders. Despite their severe consequences and high mortality rates, eating disorders are still considered a niche specialty and are often overlooked by journals (Marzola et al., 2022). These challenges in research dissemination may have serious implications for clinical advancement (Marzola et al., 2022).

This study presents a comprehensive meta-analysis examining cytokine levels in individuals with eating disorders (EDs) compared to healthy controls, with particular attention to subtype differences and longitudinal changes across time points. A total of 43 studies were included. In anorexia nervosa (AN) compared to healthy controls (HCs), IL-6 and IL-15 levels were higher, while IL-7 levels were lower. No differences were found for TNF- α , IL-1 β , IL-4, IL-8, IL-10, IFN- γ , MCP, and TGF- β when controlling for outliers. Longitudinally, IL-6 levels were lower at follow-up compared to baseline in AN, while TNF- α and IL-1 β remained unchanged. No differences emerged based on AN subtype. Individuals with bulimia nervosa (BN) showed no differences in IL-6 and TNF- α compared to HCs. The available data were insufficient to conduct meta-analyses for binge-eating disorder (BED) or other EDs.

This is a well-conducted meta-analysis of a large number of studies, offering a robust overview of cytokine alterations in EDs. The statistical methods employed are both innovative and appropriate, enhancing the reliability and validity of the findings. The Introduction clearly outlines the need for this research and provides a comprehensive overview of immunology in EDs. The Discussion thoughtfully contextualizes the findings within the broader literature, offering a clear interpretation of their clinical and theoretical implications.

We thank you wholeheartedly for your positive comments and for your thorough review.

I suggest the following minor points for consideration, listed in order of appearance, as hints for improving the manuscript:

Abstract:

1. The full name of “IFN- γ ” should be provided when first mentioned.

Thank you for pointing this out. This has been amended.

2. The text states: “Conclusion: In acute AN, concentrations of IL-6 and IL-15 are elevated and IL-7 is decreased, with evidence for normalisation of IL-6 over the course of weight restoration.” The findings, as explained in the Results, show a reduction with a small effect size; however, there is no mention of “normalisation”. Accordance should be reached.

This has been amended to “...with preliminary but inconclusive evidence for small decreases in IL-6 over the course of weight restoration”.

3. Keywords should be added at the end of the Abstract.

We have added keywords: “Keywords: Anorexia nervosa; bulimia nervosa; cytokines; eating disorders; meta-analysis.”

Materials and Methods:

4. Page 6: The adherence to the PRISMA checklist and the quality assessment should be mentioned in the Abstract, if word count allows.

We have added this to the methods section of the abstract: “This systematic review and meta-analysis **following PRISMA guidelines** assessed cross-sectional and longitudinal studies from four databases (PubMed, Web of Science, MEDLINE and PsycINFO) reporting cytokine concentrations in people with EDs. Random-effects models were utilised for all meta-analyses.”

5. Page 7, section “2.3. Source Selection”: The sentence “The search process was conducted independently by two reviewers (JLK and CB)” should be complemented with information on how discrepancies were resolved.

This sentence has been modified to: “The search process was conducted independently by two reviewers (JLK and CB) **and discrepancies in the eligibility assessment were discussed and resolved through discussion including a third reviewer (BD).**”

Results:

6. In Table 2, I recommend adding a column indicating whether the studies controlled for

clinical and lifestyle confounders that may affect cytokine levels, specifying which variables were considered.

Thank you for your valuable suggestion. As there was a considerable amount of information to add to the table, we decided to add a separate table that provides a detailed overview of the variables controlled for by studies in the supplementary materials (Table S2). We have signposted to this table in Section 3.2. Quality assessment of included studies: “A detailed overview of the variables controlled for in individual studies is provided in Table S2.”

7. Page 20, end of section “3.1. Characteristics of included studies”: “CBT” appears twice, which may be unintended.

This was intentional; the sentence before was referring to that there was a) weight gain treatment with CBT, CBT alone, and CBT with pharmacological treatment. We have tried to make this clearer with the following amendment: “ The intervention used across studies included inpatient weight restoration treatment (n=5), specialist ED treatment (n=1), unspecified weight gain treatment (n=1) and with cognitive behavioural therapy (CBT; n=1), CBT **alone** (n=1), and CBT with pharmacological therapy (n=1). In one study, treatment was unspecified.”

8. In the sections “3.3.1.1. Tumour-necrosis factor- α (TNF- α)” and “3.3.1.2. Interleukin-6 (IL-6)”, the number of participants included in the AN subtype analysis should be reported.

This information has now been added:

“A total of 15 studies reported AN subtype data for TNF- α , which indicated no subgroup difference between the AN-R (**AN-R n = 264; HC n = 315**) and AN-BP (**AN-BP n = 40; HC n = 182**) **meta-analyses** ($p = 0.690$; Figure S25).”

“A total of 15 studies reported AN subtype data, which indicated no subgroup difference between the AN-R (**AN-R n = 370; HC n = 433**) and AN-BP (**AN-BP n = 36; HC n = 180**) **meta-analyses** ($p = 0.790$; Figure S26).”

9. In the sections “3.3.1.4. Other cytokines available for meta-analysis” and “3.3.2. Bulimia nervosa”, information regarding the number of studies and total participants should be included.

This information has been added to section 3.3.2. Bulimia nervosa:

“3.3.2.1. *Interleukin-6 (IL-6)*

The meta-analyses of the studies (**n = 5**) in BN populations indicated higher concentrations of IL-6 in participants with BN (**n = 163**) compared with HCs (**n = 115**), although this was not significant (Table 3; Figure 5). One study (37) was deemed an outlier, which is likely due to the higher mean BMI of the sample (41.1kg/m²). Removing this study from the IL-6 meta-analysis reduced the pooled SMD to 0.30 (95% CI -0.25, 0.86; *p* = 0.287; *I*² = 65%).

3.3.2.2. *Tumour necrosis factor-α (TNF-α)*

There were significantly higher concentrations of TNF-α in participants with BN compared with HCs (Table 3; Figure 5) **across five studies (BN n = 145, HC n = 123)**. The heterogeneity estimates were considerably high for both analyses. The funnel plot indicated asymmetry (Figure S17) which was confirmed by a significant Egger's test for small study effects (*z* = 4.91, *p* < 0.001). The same study (37) was deemed an outlier and removing this study reduced the overall effect size (SMD = 0.72) rendering the difference between groups non-significant (95% CI -0.14, 1.58; *p* = 0.101) and slightly reducing the *I*² heterogeneity statistic to 84%. After removing this study the Egger's test was no longer significant (*z* = 1.30, *p* = 0.192).

For section 3.3.1.4. Other cytokines available for meta-analysis, it was decided by the research team not to include this information in-text as to avoid making the text overly cumbersome to read (see comment 4 from Peer Reviewer 3). Please note that this information is presented in Table 3.

10. In the section "3.4. Longitudinal meta-analyses in AN", it is stated: "As sensitivity analyses, studies were removed from analyses where either ≤10% weight gain occurred, where participants didn't reach at least 80% ideal body weight, or where the follow-up group mean BMI was ≤17kg/m², depending on what data was reported in the study." Considering the global worsening of the clinical picture in cases of severe-enduring eating disorders (SE-ED) (Wonderlich et al., 2020), a complementary analysis focusing on time points without BMI improvement would have been valuable. If not feasible due to a lack of studies, this should be acknowledged as a limitation of the current literature, suggesting future research in this direction.

Thank you for this comment. Please note that none of these studies were conducted over a sufficiently long period of time to constitute SE-ED therefore we are unable to comment on SE-ED in this manuscript, although it is an important gap in the literature. As the reviewer is likely aware, there is current debate in the literature as to the exact definition of SE-ED or SE-AN. Once this definition has been clarified, future studies examining cytokine levels

should consider this as a subgroup of AN to examine immunological functioning in cases of prolonged illness.

The decision to use the time-point for the main analyses constituting weight restoration or discharge from care was made as only one study had a longer-term follow-up time-point at 12-months; therefore to try and establish some homogeneity it was decided to use an earlier time-point for this study. However we have added the following sentence to the limitations section:

“There was also a paucity of longitudinal studies reporting cytokine concentrations after long-term follow-ups or prospectively over the longer-term course of the illness, meaning that it was not possible to examine trajectories over illness stages (e.g., in the case of longstanding, protracted, or “severe-enduring” illness).”

11. In the sections “3.4.1. Tumour necrosis factor- α (TNF- α)”, “3.4.2. Interleukin-6 (IL-6)”, and “3.4.3. Interleukin-1 β ”, the time span between assessments should be reported. This information should also be addressed in the Discussion and Abstract. The same applies to the change in BMI between time points (Δ BMI), if available.

The time in-between assessments and mean change in BMI has now been added to the results section:

“3.4.1. Tumour necrosis factor- α (TNF- α)

The meta-analysis of TNF- α included 267 AN participants at baseline and 240 at follow-up (**mean BMI increase 2.85kg/m², reported by eight studies**) from nine studies, finding no significant difference between time-points (SMD = -0.05, 95% CI -0.22, 0.13; $p = 0.617$ $I^2 = 0\%$; Table 4; Figure S29). **The follow-up period across studies ranged from 7.3 to 32.2 weeks (M \pm SD = 14.1 \pm 8.3 weeks).** The trim and fill procedure imputed two missing studies (Figure S23), increasing the SMD to -0.07 (95% CI -0.24, 0.10). The estimated SMD remained similar when removing three studies (28, 66, 70) where weight increase was insufficient (SMD = -0.04; 95% CI -0.23, 0.15; $z = -0.43$; $p = 0.670$).

3.4.2. Interleukin-6 (IL-6)

Across eight studies with 279 AN participants at baseline and 239 participants at follow-up collectively (**mean BMI increase 2.87kg/m², reported by six studies**), concentrations of IL-6 were significantly higher at baseline than follow-up with a small effect size (SMD = 0.21; 95% CI 0.01, 0.42; $p = 0.042$; $I^2 = 22\%$; Table 4; Figure S30). **The follow-up period across studies ranged from 7.3 to 52 weeks (M \pm SD = 16.8 \pm 14.7 weeks).** The Egger’s test for small study effects was significant (Table 4) and the trim and fill procedure identified three

missing studies (Figure S24). When imputing these studies, the re-estimated SMD lowered to 0.10 and was non-significant (95% CI -0.13, 0.32). Additionally, when removing two studies where the BMI at follow-up was insufficient (28, 70), the SMD reduced to 0.19 and again was non-significant (95% CI -0.05, 0.43; $z = 1.52$; $p = 0.129$).

3.4.3. Interleukin-1 β

Five studies with 125 AN participants at baseline and 104 participants at follow-up (**mean BMI increase 2.40kg/m², reported by four studies**) were available for the meta-analysis of IL-1 β concentrations. **The follow-up period across studies ranged from 9.7 to 19.6 weeks (M \pm SD = 13.3 \pm 4.3 weeks)**. There was no significant difference between the baseline and follow-up time-points in IL-1 β concentrations (SMD = 0.003; 95% CI -0.37, 0.37; $p = 0.988$; $I^2 = 39\%$; Table 4; Figure S31). After removing two studies where there was insufficient weight increase (28, 70), the SMD increased to -0.10 but remained non-significant (95% CI -0.68, 0.48; $z = -0.34$; $p = 0.733$).

We have also added a new paragraph in the discussion to this effect:

“Across the longitudinal studies, the average BMI increase ranged from 2.40kg/m² to 2.87kg/m², although in approximately half of studies in each analysis, the follow-up BMI did not surpass 18.5kg/m². Therefore, it is possible that changes from baseline to follow-up may have been more pronounced with more consistent weight restoration. With further longitudinal studies including multiple time-points, future meta-analyses could explore the temporal dynamics of cytokine concentrations according to different BMI thresholds. Additionally, there was considerable variability in the time interval between baseline and follow-up (7.3 to 52 weeks across all studies), which may have contributed to our findings. The current evidence may signal that there is an absence of a substantial change in cytokine levels longitudinally, although more research is required.”

Discussion:

1. A further limitation of the existing literature – and a potential explanation for inconsistent findings – is that cytokine levels are influenced by circadian rhythms and may vary depending on the time of sampling (Nilsson et al., 2016; Zeng et al., 2024; Zielinski and Gibbons, 2022). I would suggest that the authors explicitly acknowledge this as a limitation, noting that the studies included did not control for time of blood collection. Additionally, I suggest recommending in Table 5 greater standardization on this point in future research.

Thank you for highlighting this important point that was missed. We have added “time of sampling (circadian rhythms)” and the suggested references to the text on page 34, and have added this also to Table 5 with the description/recommendation “Report the time at

which the sample was collected, and preferably collect the sample at consistent times across participants and time-points (e.g., in the morning following an overnight fast).”

References:

- Marzola, E., Panero, M., Longo, P., Martini, M., Fernández-Aranda, F., Kaye, W.H., Abbate-Daga, G., 2022. Research in eating disorders: the misunderstanding of supposing serious mental illnesses as a niche specialty. *Eating and Weight Disorders - Studies on Anorexia, Bulimia and Obesity* 27, 3005–3016. <https://doi.org/10.1007/s40519-022-01473-9>
- Nilsson, G., Lekander, M., Åkerstedt, T., Axelsson, J., Ingre, M., 2016. Diurnal Variation of Circulating Interleukin-6 in Humans: A Meta-Analysis. *PLoS One* 11, e0165799. <https://doi.org/10.1371/journal.pone.0165799>
- Wonderlich, S.A., Bulik, C.M., Schmidt, U., Steiger, H., Hoek, H.W., 2020. Severe and enduring anorexia nervosa: Update and observations about the current clinical reality. *International Journal of Eating Disorders* 53, 1303–1312. <https://doi.org/10.1002/eat.23283>
- Zeng, Y., Guo, Z., Wu, M., Chen, F., Chen, L., 2024. Circadian rhythm regulates the function of immune cells and participates in the development of tumors. *Cell Death Discov* 10, 199. <https://doi.org/10.1038/s41420-024-01960-1>
- Zielinski, M.R., Gibbons, A.J., 2022. Neuroinflammation, Sleep, and Circadian Rhythms. *Front Cell Infect Microbiol* 12. <https://doi.org/10.3389/fcimb.2022.853096>

Reviewer #2

This systematic review and meta-analysis represents a valuable expansion of prior work on cytokine profiles in eating disorders, incorporating 24 additional studies (total N=43) to examine cytokine differences between anorexia nervosa (AN), bulimia nervosa (BN), and healthy controls, while tracking longitudinal changes during AN treatment. The key findings are:

- AN vs. HC: elevated IL-6 and IL-15, alongside reduced IL-7, with no differences in TNF- α or IL-1 β after removing outliers.
- Longitudinal AN: While IL-6 appeared to decrease post-treatment initially, this effect disappeared after trim-and-fill adjustment. TNF- α and IL-1 β showed no changes.
- BN vs. HC: No significant cytokine differences emerged after outlier adjustment.

We thank you wholeheartedly for your comments and review.

There are some major issues, listed below:

- Results (3.4.2) and the discussion. Looking at Figure S24, I notice significant funnel plot asymmetry in your longitudinal IL-6 analysis. The trim-and-fill procedure clearly shows that imputed studies reduce the effect to non-significance, yet the main text doesn't adequately address this critical finding. I would advise revising your conclusions to acknowledge that the IL-6 "decrease with weight restoration" may be an artifact of publication bias. Consider discussing how selective reporting of positive longitudinal results could explain this pattern.

We thank the reviewer for pointing this out and for carefully reviewing our figures. Whilst we have highlighted this already in the results, we have added this possible explanation to the discussion and abstract:

Results of abstract: "Longitudinally, IL-6 was lower in AN at follow-up compared to baseline, **although this may be an artefact of publication bias**. TNF- α and IL-1 β did not change longitudinally."

Conclusion of abstract: "In acute AN, concentrations of IL-6 and IL-15 are elevated and IL-7 is decreased, with **preliminary but inconclusive evidence** for small decreases in IL-6 over the course of weight restoration."

Discussions: "Longitudinally, meta-analyses were performed to examine differences in concentrations of IL-6, TNF- α and IL-1 β from baseline to follow-up in AN, finding significant but small decreases over time only for IL-6, **although this may be due to publication bias**

as this finding was no longer significant when adjusting for funnel plot asymmetry.”

(paragraph 1)

“Overall, it can be concluded that increases in IL-6 in the acute stages of AN are more robust than increases in TNF- α , and increases in IL-15 and decreases in IL-7 in AN are a new finding.

Concentrations of IL-6 decrease longitudinally in people with AN, **which may be associated with weight recovery, although this may also be due to publication bias and thus needs further investigation.**” (paragraph 1 under 4.2. Conclusions and future directions).

- Results (3.1). While I see you've included medication use in your meta-regressions (Table S2), the null result ($B=-0.01$, $p=0.202$) doesn't necessarily rule out confounding. For instance, Käver et al. (2024) reported 45% SSRI use, and SSRIs can independently modulate cytokines why I would encourage conducting sensitivity analyses excluding studies with >20% medicated participants. This would strengthen your conclusions.

Thank you for this fantastic suggestion which has significantly improved the manuscript. Interestingly, we did not find that the percentage of the sample using psychotropic medication was a significant moderator, although excluding these studies did lead to an increase in SMD across all main analyses in AN vs HC (IL-6, TNF- α , IL-1 β). We decided on a conservative threshold of >10% of the sample.

The findings changed as follows (new addition to manuscript, inserted in relevant sections):

TNF- α : “When removing studies where >10% of the sample was using psychotropic medication (11, 12, 39, 48, 60, 65), the SMD increased to 0.41 ($p = 0.011$). Heterogeneity tests remained consistent ($I^2 = 85\%$; Egger's test $z = 0.99$, $p = 0.322$).”

IL-6: “When removing studies where >10% of the sample were using psychotropic medication (11, 12, 39, 48, 60, 65), the SMD increased to 0.50 and remained significant ($p<0.001$), although heterogeneity remained high ($I^2 = 82\%$) and the Egger's test was significant ($z = 2.33$; $p = 0.020$).”

IL-1 β : “There were three studies where >10% of the sample were using psychotropic medication (11, 60, 65). When removing these studies, the SMD increased substantially to 0.79 and became significant ($p = 0.026$), although heterogeneity estimates were similar ($I^2 = 90\%$).”

In the discussion (paragraph 1) we added: “These analyses did not differ according to AN subtype, **but when removing studies where >10% of the sample were taking**

psychotropic medication, the effect size increased for IL-6, TNF- α and IL-1 β (the latter becoming significant, with elevated levels in AN)."

In section 4.2. Conclusions and future directions we added the following:

"The present study found an increase in differences between AN and HC in IL-6, TNF- α and IL-1 β when samples with a high usage of psychotropic medication were excluded, likely as medications such as selective serotonin reuptake inhibitor (SSRIs) usage is associated with lower levels of IL-6, TNF- α and IL-1 β (92). Therefore it is particularly important that studies report data according to whether individuals are or are not taking psychotropic medication, or consider this as an exclusion criterion."

- Results (3.3.1.2). Your MetaForest analysis (Figure S34) primarily identified regional differences (North America vs. Europe) rather than biologically meaningful moderators. While this is interesting, I think highlighting ML's utility may be misleading when the main finding reflects geographic heterogeneity in protocols rather than pathophysiology. I recommend reframing these results as exploratory and emphasizing that regional discrepancies likely reflect methodological differences rather than biological insights.

Thank you for this comment. We would like to emphasise that the moderators entered into the MetaForest analyses were not selected to reflect differences in biology of the samples, but also for methodological differences (i.e., the analyses were examining sources of heterogeneity which could be both methodological or biological). We have added the word "exploratory" where we introduce these analyses in the results section. Given that this finding pertaining to geographical differences was not consistent across cytokines, and most moderators were non-significant, we did not expand on this result in the discussion. Therefore we have not commented on whether this result is likely to be due to methodological or biological differences, as (as you have correctly stated) it is not possible to understand the origins of this finding. As a third point, it could be that lifestyle differences (e.g., diet) across regions could be important for this finding.

Minor issues:

- Table 1. I may have missed this but I am lacking a definition of adult and adolescent in terms of age ranges.

This has now been added:

- Adolescents (12-18 years)
- Adults (≥ 18 years)

- Mixed

- Results (3.1, Table 2). Your BN sample's mean BMI of 31.7 ± 10.4 kg/m² seems unusually high and likely includes atypical cases. Looking at Figures S16–S17, I can see extreme outliers that probably reflect this heterogeneity. I would advise either excluding these outliers or stratifying your BN analyses by BMI (e.g., typical BN [BMI 18–25] vs. atypical [BMI ≥ 30]) to avoid conflating different phenotypes.

We thank the reviewer for highlighting this point. We have looked into this further and can confirm that the reason the pooled BMI is so high, is because of the high BMI of one study (Tabasi et al., 2020), which had a sample BMI of 41.1kg/m²; the study was of a bariatric population undergoing sleeve gastrectomy.

Incidentally this study was already highlighted as an outlier in the analyses and the SMD was recalculated with its removal. We have explained this as a potential source of being an outlier in the results section: “One study (37) was deemed an outlier, **which is likely due to the higher mean BMI of the sample (41.1kg/m²).**”

We have also recalculated the average BMI excluding this study and have reported it in section 3.1. Characteristics of included studies: “The mean BMI of the AN participants was 15.7 ± 2.3 kg/m² (n=34 studies), of BN participants was 31.7 ± 10.4 kg/m² (n=4 studies; **22.0 \pm 3.1kg/m² when excluding one study with a specifically bariatric population (37)**), and of HCs was 21.2 ± 3.0 kg/m² (n=29 studies).”

- Results (3.3.1.1); Supplementary Figures S5, S7. Your TNF- α and IL-1 β funnel plots show clear asymmetry suggesting publication bias, but this isn't mentioned in the main text. Thus I suggest to briefly describing these patterns in your Results section to enhance transparency.

Thank you for highlighting this. These studies with high effect size estimates as shown in the funnel plots were identified as outliers from an inspection of the 95% CIs (as stated in the methods section). We have added in a brief description of the funnel plots in the results section, see the following:

3.3.1.1. *Tumour-necrosis factor- α (TNF- α)*

Across 30 studies with a total of 1636 participants (AN n = 872; HC n = 764), six found elevated levels of TNF- α and none reported reduced levels (Figure 2). Across all studies there was a small difference in TNF- α concentrations between AN and HCs whereby AN had elevated levels (SMD = 0.27; 95% CI 0.01, 0.54; $p = 0.040$). Heterogeneity estimates were

high ($I^2=84\%$; Table 3) and although there was no evidence of publication bias from the Egger's test, **there was clear funnel plot asymmetry with several studies showing an abnormally high SMD (Figure S5). Subsequently** six outliers were identified (12, 32, 41, 58, 61, 69). After removing these outliers, the effect size became marginal and non-significant (SMD = 0.06; 95% CI -0.10, 0.22; $z = 0.71$; $p = 0.475$) and the I^2 heterogeneity estimate reduced to 47.26%.

3.3.1.2. Interleukin-6 (IL-6)

Across 25 studies including a total of 1596 participants (AN $n = 822$, HC $n = 774$), nine exhibited elevated levels and two exhibited reduced levels of IL-6 in AN (Figure 3). The pooled mean concentrations of IL-6 were significantly higher in AN compared to HC with a small effect (SMD = 0.36; 95% CI 0.11, 0.61; $p = 0.005$; Table 3; Figure 2). This analysis showed high heterogeneity ($I^2 = 82\%$) and the Egger's test for small study effects was significant. Five studies were identified as outliers following an inspection of the 95% CIs (11, 12, 32, 41, 61) **and the funnel plot (Figure S6)**. When removing these studies the SMD was slightly lower but remained significant (SMD=0.32; $p = 0.001$); the I^2 statistic lowered to 65.1% and the Egger's test was no longer significant ($z = 1.98$, $p = 0.050$).

3.3.1.3. Interleukin-1 β (IL-1 β)

Across 14 studies with a collective sample size of 598 participants (AN $n = 354$, HC $n = 244$), there was a non-significant moderate-sized increase in pooled concentrations of IL-1 β between AN and HC groups (SMD = 0.53, 95% CI -0.06, 1.12; $p = 0.080$; $I^2 = 91\%$; Table 3; Figure 4). From exploring the heterogeneity (95% CIs) **and the funnel plot whereby two studies had abnormally high SMDs and one low (Figure S7)**, three studies were identified as outliers (41, 61, 65). The effect size was smaller and remained non-significant after removing these studies (SMD = 0.23; 95% CI -0.04, 0.50; $z = 1.67$; $p = 0.095$; $I^2 = 44\%$).

Reviewer #3

I read with much interest the article titled “ A comprehensive updated cross-sectional and longitudinal meta-analysis of cytokines in eating disorders”. This study updates a prior meta-analysis by integrating 24 new studies examining cytokine levels in people with eating disorders, with a focus on anorexia nervosa. It finds that IL-6 and IL-15 are elevated, and IL-7 is reduced in acute anorexia, while most other cytokines do not show consistent changes. The analysis also includes longitudinal data showing that IL-6 levels tend to decrease during weight restoration, suggesting possible normalization of immune function over time. However, I noted some points that should be addressed by authors:

1- The inclusion of longitudinal data and application of machine learning (MetaForest) for moderator detection are substantial advancements. However, core findings on cytokines such as IL-6 and TNF- α remain similar to earlier reviews. Please emphasize the novel insights from longitudinal findings and machine-learning-based heterogeneity analyses more clearly in the abstract and discussion to highlight added value.

Thank you for highlighting this point. The following addition was made to the abstract: “This study is an update of a previously published meta-analysis, **including longitudinal analyses and machine learning heterogeneity analyses (MetaForest).**”

When considering that the longitudinal analyses may suffer from publication bias, as highlighted by Reviewer 2, the current level of detail on these findings seems appropriate. Moreover the novel aspects of the study are highlighted in Section 4.1. Strengths and limitations: “We were able to expand on the previous meta-analysis both by (a) including new studies of cytokines that have previously been meta-analysed; (b) meta-analysing several new cytokines that have not been previously examined; (c) examining changes in pro-inflammatory cytokines longitudinally in people with AN; and (d) employing various approaches, including machine learning, to examine potential sources of heterogeneity.”

2- Strengthen the limitations section by explicitly stating the impact of female-only samples and variable control for confounders. Consider subgroup analysis where possible on medication status or BMI level. If there is no data just add to the limitations

Thank you for your recommendation. Unfortunately due to limited data reporting these subgroup analyses were not possible. We have amended the discussion to that effect:

“There was also a paucity of longitudinal studies reporting cytokine concentrations after long-term follow-ups or prospectively over the longer-term course of the illness, meaning that it was not possible to examine trajectories over illness stages (e.g., in the case of

longstanding, protracted, or “severe-enduring” illness). Likewise, we could unfortunately not perform subgroup analyses by BMI category, sex or psychotropic medication status as data were not reported in the studies. It is likely that these are important moderators given that cytokine levels fluctuate according to weight (85), menstrual cycle stage and sex (86, 87), and psychotropic medication usage (18).”

“The present study found an increase in differences between AN and HC in IL-6, TNF- α and IL-1 β when samples with a high usage of psychotropic medication were excluded, likely as medications such as selective serotonin reuptake inhibitor (SSRIs) usage is associated with lower levels of IL-6, TNF- α and IL-1 β (92). Therefore it is particularly important that studies report data according to whether individuals are or are not taking psychotropic medication, or consider this as an exclusion criterion.”

3- Rephrase language suggesting causality or normalization unless directly supported by prospective trial data. Replace with terms such as "associated decrease" or "observed reduction over time."

Thank you for pointing this out. We have removed the reference to “normalisation” and have attenuated the language in several places in the manuscript, particularly in reference to the longitudinal meta-analyses, so as not to suggest that changes in cytokine levels are caused by increases in BMI.

4- The manuscript is clearly written with strong structure. However, the results section is densely packed with data and could overwhelm readers. It would be better to move some detailed statistical figures or forest plots into supplementary materials and condense result presentation into clearer thematic sub-sections.

Achieving legibility is a challenge with such a broad and comprehensive meta-analyses. We have reviewed in detail the results section and have aimed to balance providing necessary statistical information for comprehending main analyses and sensitivity analyses, and not overwhelming readers. We have placed most forest plots, heterogeneity plots (e.g., funnel plots), moderator analyses and sensitivity analyses to the supplementary materials to try and streamline the results section. The remaining figures are included in the main manuscript as they are necessary for understanding the key findings. We have also introduced subheadings for “Sensitivity and moderator analyses” under the broader meta-analyses where sensitivity and moderator analyses were possible (i.e., IL-6, TNF- α and IL-1 β in AN).

5- Provide narrative justification when referencing prior meta-analyses or conflicting findings.

Ensure figures use consistent formatting and define abbreviations within each figure/table legend.

We have reviewed the figures and tables (including the supplementary file) to ensure consistency in formatting and when defining abbreviations. Please note that Figure 5 purposefully does not include a line of effect, as this would be misleading considering that there are two separate meta-analyses included in this plot (they were combined into one plot to simplify the manuscript).

6- Define each acronym at first mention in the abstract and main text. Add a glossary or abbreviation list if space permits.

We have reviewed the manuscript to ensure that each acronym is defined at first use. Unfortunately there is insufficient space for a glossary.

Final Revision Instructions

*To the Author— Please review the editorial comments and requests below and confirm that changes have been made in the manuscript in the right-hand column. The completed document **must be uploaded** as a related manuscript file.*

Files and General Policies	
Main manuscript file must be in Microsoft Word or LaTeX format. LaTeX and Tex article source files must be accompanied by the compiled PDF for reference. The bibliography must be submitted separately (as a .bib file) or contained within the .tex file.	Document is in word format.
Each Figure must be provided as a separate file and must be supplied whole, with all panels included in a single document. Figures should be provided at a minimum resolution of 300 dpi at final size. Figure files must only contain images (please also leave out labels such as “Figure 1” etc). Figure captions must instead be included within the main manuscript file, grouped together at the end of the document.	This has been amended.
All figures, tables, and supplementary items must be cited in the manuscript and numbered in the order in which they appear.	This has been amended, particularly in reference to the supplementary materials which have been reformatted to ensure that the files are in order of how they have been cited.

Please check whether your manuscript contains third-party images, such as figures from the literature, stock photos, clip art or commercial satellite and map data. We strongly discourage the use or adaptation of previously published images, but if this is unavoidable, please request the necessary rights documentation to re-use such material from the relevant copyright holders and return this to us when you submit your revised manuscript. An appropriate permissions statement must be present in the relative figure caption for any third-party images.	There are no third-party images.
Tables must be provided in an editable format and should be grouped together at the end of the main manuscript file.	The tables have been moved to the end of the file.
The reporting summary will be published alongside your manuscript therefore it needs to accurately represent your work. In this case, please take a closer look at the reporting summary and make sure things are completed correctly, and upload this with your revised manuscript. If an item does not apply, for example human participants, I need you to check the NA box next to that item. No section should be left blank. Also, please make sure to include your name and date at the top of the document. If you require a new Reporting Summary form, please download it here: https://www.nature.com/documents/nr-reporting-summary.zip	This has been recompleted and checked.
Your paper will be accompanied by a two-sentence summary when it is published on our homepage. Please provide a draft summary below (up to 350 characters max. including spaces, in the present tense) using the examples on our website as a guide (beneath each paper's title). Keeler et al.	Keeler et al. report an updated meta-analysis of peripheral cytokine concentrations in individuals with eating disorders. Results indicate elevations in interleukin (IL)-6 and IL-15 and reductions in IL-7 in people with anorexia nervosa in comparison to healthy controls, and no elevations in pro-inflammatory cytokines in people with bulimia nervosa.

ORCID

Communications Medicine is committed to improving transparency in authorship. As part of our efforts in this direction, we are now requesting that all authors identified as 'corresponding author' create and link their Open Researcher and Contributor Identifier (ORCID) with their account on the Manuscript Tracking System (MTS) prior to acceptance. ORCID helps the scientific community achieve unambiguous attribution of all scholarly contributions. For more information please visit <http://www.springernature.com/orcid>.

For all corresponding authors listed on the manuscript, please follow the instructions in the link below to link your ORCID to your account on our MTS before submitting the final version of the manuscript. If you do not yet have an ORCID you will be able to create one in minutes.

IMPORTANT: All authors identified as 'corresponding author' on the manuscript must follow these instructions. Non-corresponding authors do not have to link their ORCIDs but are encouraged to do so. Please note that it will not be possible to add/modify ORCIDs at proof. Thus, if they wish to have their ORCID added to the paper they must also follow the above procedure prior to acceptance.

To support ORCID's aims, we only allow a single ORCID identifier to be attached to one account. If you have any issues attaching an ORCID identifier to your MTS account, please contact the Platform Support Helpdesk at <http://platformsupport.nature.com/>

I can confirm that my ORCID ID is linked (Johanna Keeler, corresponding author).

We regularly highlight papers published in Communications Medicine on the journal's Social Media accounts (e.g. Bluesky: commsmed.nature.com). If you would like us to mention authors, institutions, or lab groups in these posts, please provide the social media along with the respective handles in the right-hand column.	Bluesky tag: @jhnnklr @kingsedresearch
Please let us know if there are any plans for an Institutional press release on your article.	There are no current plans.
We would welcome the submission of material for the 'Featured Image' section on the Communications Medicine home page. Images should relate to the content of your manuscript but need not be contained within the paper. Photographs and aesthetically interesting images are preferred; diagrams are generally not used. Suggestions should be uploaded as a Related Manuscript file. Please provide 1200x675-pixel RGB images. You will also need to submit a completed Image License to Publish. Unfortunately, we cannot promise that your suggestions will be used.	N/A
Supplementary information	
All Supplementary Figures, Tables, Methods, Notes, Discussion and References should be organized into a single PDF (excluding any non-flat files) named Supplementary Information.	The file is a PDF titled Supplementary Information
Where Supplementary Data files are provided, each file must be labelled as Supplementary Data 1, etc., and legends for these must be given in the column to the right (not in the Supplementary Information file). Each item must also be cited in the main manuscript using the same formatting style.	Supplementary Data 1. Cytokine data used for meta-analyses.
Large datasets must be supplied as separate Supplementary Data files. Each file must be labelled as Supplementary Data 1, etc., and legends for these	N/A

must be given here in the column to the right (not in the Supplementary Information file).	
All source data, i.e. the numerical results underlying the graphs and charts presented in the main figures, must be made available as Supplementary Data (in Excel or text format) or via a generalist repository (eg, Figshare or Dryad). Please include a caption for the source data file in the column to the right, and add a sentence to the Data Availability statement to describe how source data can be accessed using text similar to “The source data for Figure X is in Supplementary Data Y”. It needs to be specific about which figures you provide source data for, and specifically where it can be found.	The legend has been included above. The data availability statement has been amended as requested to: “The source data for Figures 2, 3 and 4 are in Supplementary Data 1 (sheets 1, 2 and 3, respectively). The source data for Figure 5 is in Supplementary Data 1, sheet 5. The source data for Figures S30 and S31 is in Supplementary Data 1, sheet 4. The source data for Figures S34, S35 and S36 is in Supplementary Data (sheets 6, 7 and 8, respectively).”
Supplementary Information files will be uploaded with the published article as they are submitted with the final version of your manuscript. Any highlighting or tracked changes should be removed from the file.	This has been actioned.
Please ensure that the title and author list given on the cover page of your Supplementary Information matches the final title and author list given in the main manuscript file.	This is identical.
Supplementary items must be cited in a consistent format throughout the Supplementary Information, and in line with the citations used in the main manuscript. We recommend using the following formats: Supplementary Figure 1, Supplementary Table 1, Supplementary Data 1, Supplementary Note 1, Supplementary Discussion, and Supplementary References.	The supplementary items are cited in a consistent format and consistent with how they are referred to in-text.
Each Supplementary item (Figure, Table, Note, etc.) must be individually cited in the main manuscript. These can be cited in blocks, e.g. “for example images see Supplementary Figures XX-YY”.	Each supplementary item is cited in the main manuscript.

Supplementary References should appear at the end of the Supplementary Information file and should be self-contained and numbered from 1. References mentioned in both the main text and the Supplementary Information must be part of both reference lists so that the Supplementary Information does not refer to the reference list in the main paper and vice versa. Please check that your Supplementary References comply with this style.	This has been checked, and there are separate reference lists per file.
Title Page	
Please ensure that the author list provided in our manuscript tracking system matches the author list in the main manuscript.	They match
Please check that your author list and affiliations comply with the following: We recommend that author first names be written out in full rather than provided as initials. Titles (Professor, Dr, etc.) and qualifications (PhD, MD, etc.) must not be included. Affiliations must be numbered in the order of their first appearance in the author list. Each affiliation must contain only one address. Each affiliation must include the institution, city and country. The name of the city must be provided separately from the institution even if it is a part	This has been checked.

of the institution name; e.g. 'University of Science and Technology Beijing, Beijing, China' Author tagging statements are limited to the following two options: "These authors contributed equally" and "These authors jointly supervised this work", with no more than one of each tag permitted. Where relevant, "present address" must be provided separately as the final affiliation. The number of equally contributing authors must be limited to six. The number of jointly supervising authors must be limited to six. The number of corresponding authors must be limited to four. At least one corresponding author must be designated, and an e-mail address must be provided for each corresponding author (with a limit of one e-mail address per author).	
Please note that titles should be a single declarative sentence of ~15 words. We recommend the following title and ask that you update the title in the manuscript file and submission system: Cytokine concentrations in people with eating disorders: A systematic review and meta-analysis Or	We have updated this however feel that it is important to reflect that this is an updated review; therefore it has been changed to: "Cytokine concentrations in people with eating disorders: A comprehensive updated systematic review and meta-analysis"

Cytokines and people with eating disorders: A systematic review and meta-analysis	
The abstract should be accessible to non-specialists and be up to 250 words long. It should contain four subsections with the following section headings: Background, Methods, Results, and Conclusions. The ‘Background’ should provide the context, rationale and aim of the study; the ‘Methods’ should briefly describe how the study was performed including details on the sample/cohort studied, any key methods and statistical methods; the ‘Results’ should start with “Here we show” or an equivalent phrase, and briefly describe the main findings; and the ‘Conclusions’ should provide a brief summary of the study and discuss potential implications. The Results and Conclusions of the paper should be written in the present tense. The abstract must not contain references and acronyms should be avoided. We recommend the following edits to your abstract: Add the OSF registration link/number in the Methods part of the Abstract. Also, add the dates of the literature search.	The recommended additions have been included.
A Plain Language Summary should be placed immediately after the Abstract. The aim of the Plain Language Summary is to make the research accessible to members of the general public to allow them to engage with medical research, which increases understanding and can help build trust. The Plain Language Summary should be written in simple and clear language, avoiding specialist jargon and long sentences, making it easy to	This has been added.

read and understand by the general public with no scientific training, including non-native speakers of English.

A summary should focus on the study as a whole, capturing its essence without focusing on details, making it clear why the research was done and why the findings are significant and relevant to the reader.

A Plain Language Summary should be around 120 words long. We would recommend the following structure:

- briefly explain the aim of the study and its background / context;
- briefly describe how the research question was approached (methods);
- briefly describe the main finding and take-home message of the study;
- briefly discuss the significance of this finding and potential future implications for members of the general public.

You can also use the following examples for guidance:

<https://www.nature.com/articles/s43856-021-00038-8>

<https://www.nature.com/articles/s43856-021-00013-3>

Main text

To comply with our article templates, the text must be split into the following sections (in this order):

We have verified that the article is split into these sections. We have removed the subheading “4.1. Strengths and limitations” in

- “Introduction”, which must include the background and rationale for the work. The final paragraph should be a brief summary of the major results and conclusions in the present tense. The results of the current study must only be discussed in this final paragraph. The Introduction should contain no references to figures or tables. - Methods, which should be split into subheaded sections. - “Results” or “Results and Discussion”, which should be split into subheaded sections. Figures should not be embedded in the text but submitted separately. - Discussion (optional, only if not provided with the results), without subheadings. To improve readability, we recommend that the main text (Introduction, Results and Discussion) be limited to ~5000 words.	the discussion however have kept the conclusions subheading as this was requested below.
The discussion should consist of an extended analysis of the results and their comparison to the literature rather than a short summary or conclusion. Please integrate your summary paragraph at the end of the Discussion or add a Conclusion heading.	This has been verified. There is already a subheading for the conclusion so this has not been amended.
General citations to the Supplementary Information (e.g. "see Supplementary Information") should be avoided. Please update all such citations to specific Supplementary Items instead.	This has been amended in the main manuscript.
Please ensure that all equations are supplied in an editable format upon resubmission. Equations must be numbered sequentially.	This has been verified.
Please check that you have not copied any text directly from published work (even your own) without clear attribution, including one or more references. We run a plagiarism detection software and may need to request additional changes if we identify large blocks of identical text.	This has been verified.

Please include exact p-values where possible. We ask that you also include the name of the statistical test and the estimated effect size. If applicable, please also include the confidence interval.	This has been checked.
Avoid the use of the word “significant” unless referring to the results of a statistical test.	This has been checked.
Language such as “new”, “novel”, “for the first time”, “unprecedented”, etc, should be avoided, or qualified with “to the best of our knowledge” or similar, because it often leads to unproductive controversy. Novelty should be made clear from the context.	This language has been attenuated where applicable. Where “new” has been retained in the manuscript, this either refers to studies included since the last publication, and elsewhere is qualified by “to the best of our knowledge”. References to “novel” and “for the first time” have been removed entirely.
Our style does not allow for the use of bullet-points/lists, bold or italics to convey emphasis. Please remove such usage from your manuscript, and do not replace the use of bold/italics with speech/quotation marks.	Bullet points were previously included at the end of the Introduction; these have been removed and the list has been reformatted as a paragraph.
Use of speech marks around words or phrases should be avoided; if a phrase is non-standard, please explain the meaning instead; otherwise they are usually unnecessary.	Speech marks have been removed from severe-enduring; however please note that they were originally included as this term is disputed by people with lived experience despite being a term adopted frequently in the literature. The speech marks were included to have sensitivity to this issue.
Display items	
Wherever statistics have been derived (e.g. error bars, box plots, statistical significance) the legend needs to provide and define the n number (i.e. the	The n number for the samples are included in the figure titles.

sample size used to derive statistics) as a precise value (not a range), using the wording “n=X biologically independent samples/animals/independent experiments” etc. as applicable.	
Please check that your Figure captions comply with the following: Figures must have both a title that will appear above the Figure, and a legend that will appear below the Figure (see e.g. https://www.nature.com/articles/s42003-020-1059-1/figures/1) The Figure title must describe the Figure as a whole and must not contain reference to specific figure panels. The Figure legend must refer to and describe all panels. Abbreviations, symbols, colors, and shading present in the Figure must be defined. Please write out the symbols/colors in words (blue circles, red dashed line, etc.) within these definitions. All figure panels must be labelled using lower case letters. Please refrain from referring to sections of figures as top/bottom/left/right/, etc.	The figure titles have been reformatted to indicate the title above the figure and the legend below. There are no figure panels.
Axis and panel labels will be published as received. We recommend using a sans-serif font such as Arial or Helvetica.	We would like the figures to be published as submitted.
Please define the error bars in each Figure and Supplementary Figure where they are used. One statement at the end of each Figure caption is sufficient if the error bars are equivalent throughout the Figure.	The error bars have been defined in each figure caption.
Please check that your Tables comply with the following: Our style does not support the use of colors or shading in Tables. All Tables must contain black and white text only.	This has been checked and the tables have been modified.

Any bold/italic formatting must be either removed or defined clearly in a Table footnote. Where Tables contain images, each image should appear in its own cell in the absence of any text. All Tables must have a brief title.	
Please pay close attention to our Digital Image Integrity Guidelines. Also ensure that you retain unprocessed data and metadata files after publication, ideally archiving data in perpetuity, as these may be requested during the peer review and production process or after publication if any issues arise.	This has been confirmed.
Methods	
Please ensure that all information present in the Reporting Summary is also in the manuscript.	All information from the reporting summary is included in the manuscript.
We allow unlimited space for Methods. The Methods must contain sufficient detail such that the work could be repeated. It is preferable that all key methods be included in the main manuscript, rather than in the Supplementary Information.	All key methods are included in the main manuscript. We have added detail on the MetaForest moderator analyses from the SM into the main manuscript.
Please avoid use of “as described previously” or similar, and instead detail the specific methods used herein.	Reference to the previous study in this way has been removed from the manuscript.
No more than two layers of subheadings are permitted in the Methods. Please reformat this section accordingly.	This has been reformatted; the third layer numbering has been removed.
The Methods should include a separate section titled “Statistics and Reproducibility” with general information on how the statistical analyses of the data were conducted, and general information on the reproducibility of	We have renamed “Statistical analysis” to “Statistics and reproducibility” as this information is already included in this section.

experiments, including the sample sizes and number of replicates and how replicates were defined.	This article does not include an experiment.
Please follow the PRISMA guidelines for reporting. Please fully complete a PRISMA checklist (adding information to the manuscript where needed) and upload this with your revised manuscript. https://prisma-statement.org/PRISMAStatement/Checklist.aspx	We have appended a completed PRISMA checklist.
Data Policies	
Please move your Data Availability statement to a section titled "Data Availability" immediately below the Discussion section.	This has been moved.
Please ensure that your Data Availability statement:  ● describes how any existing datasets used in the study can be accessed; ● includes accession details for deposited data; ● mentions where source data for the figures can be found; ● and states that all other data are available from the corresponding author (or other sources, as applicable) on reasonable request. See here for more information about formatting your Data Availability Statement: http://www.springernature.com/gp/authors/research-data-policy/data-availability-statements/12330880	The data availability statement has been modified to: “The source data for Figures 2, 3 and 4 are in Supplementary Data 1 (sheets 1, 2 and 3, respectively). The source data for Figure 5 is in Supplementary Data 1, sheet 5. The source data for Figures S30 and S31 is in Supplementary Data 1, sheet 4. The source data for Figures S34, S35 and S36 is in Supplementary Data (sheets 6, 7 and 8, respectively). All other data are available from the corresponding author on reasonable request.”

Authors should cite (within the main reference list) any datasets stored in external repositories that are mentioned within their manuscript. For previously published datasets, we ask authors to cite both the related research articles and the datasets themselves. For more information on how to cite datasets in submitted manuscripts, please see our data availability statements and data citations policy.	
End Notes	
Please check that your bibliography complies with the following:  ● Your bibliography should start with the heading “References”. The references must be numbered in the order of appearance in the text, then tables, then figures. ● Any in-text citations to references (e.g. "Gupta et al. show...") should be followed by their corresponding reference citation number from the reference list. ● Manuscript citations must include journal title, article title, volume number, page or article number or DOI, and year of publication. ● No publication can be present more than once in the reference list. ● No footnotes are permitted in the references or elsewhere. Text should be incorporated into the main text, the Methods section, or the Supplementary Information instead. ● Websites should only be listed in the references if they are in common use or curated. ● Where possible, preprints in the reference list should be updated with details of the published, peer-reviewed paper. ● Citations should be formatted in the text using superscript numbers. 	This has been verified.
Please check that your 'Author Contributions' section individually lists the specific contribution of each author to the work. Each author must be	This has been verified.

referred to by name or initials. Where multiple authors possess identical initials, they must be clearly disambiguated from one another.

See our author contributions policy for further information:

<https://www.nature.com/nature-research/editorial-policies/authorship#author-contribution-statements>

File checklist					
Item	Permissible file format	File name on manuscript tracking system	File type on manuscript tracking system	Notes	
Editorial Requests Table	.doc, .docx	Editorial Requests Table	Related Manuscript File	Please provide a copy of the Editorial Requests Table supplied with our decision letter, with all changes made in response to our requests detailed in the right-hand column.	X
Cover letter (optional)	.doc, .docx, .pdf	Cover letter	Author Cover Letter	Outline any additional changes to the manuscript.	N/A
Author responses	.doc, .docx, .pdf	Response to Referees	Rebuttal Letter	Provide your point-by-point response to any issues raised by our reviewers (please include the reviewers' comments in this document).	N/A
Article File	.doc, .docx, .tex	Article File	Article (NOT revised manuscript - marked up)	Main manuscript file must be in Microsoft Word or LaTeX format. LaTeX and Tex article source files must be accompanied by the compiled PDF for reference. The bibliography must be submitted separately (as a .bib file) or contained within the .tex file.	X

Main Figure File(s)	.psd, .ai, .eps, .tiff, .jpg, .pdf, .ps, .gif, .ppt, .pptx, .png, .bmp, .vsd, .cdx, .svg or .emf	Figure 1, Figure 2, etc.	Figure	Each Figure must be provided as a separate file at a minimum resolution of 300 dpi at final size. Figures must be supplied whole, with all panels included in a single document. Figures appear at 9 or 18 cm width (1 or 2 columns respectively). Captions must not be included in the Figure files. Figure captions must instead be included within the main manuscript file, grouped together at the end of the document. Figures must be in file type .psd, .ai, .eps, .tiff, .jpg, .pdf, .ps, .gif, .ppt, .pptx, .png, .bmp, .vsd, .cdx, .svg or .emf. We recommend using vectographic formats as these lead to higher resolution figures. We strongly discourage the use or adaptation of previously published images (including figures from the literature, stock photos, clip art or commercial satellite and map data), but if this is unavoidable, you must request the necessary rights documentation to re-use such material from the relevant copyright holders and submit this to us alongside your manuscript. An appropriate permissions statement must be present in the relative figure caption for any third-party images. If individuals are identifiable in images, their written permission must be provided.	X
Main Table(s)	.doc, .docx, .tex	Article File	Article	Included within the main Article File in word-editable format.	X

				Tables should be grouped together at the end of the main manuscript file.	
Boxes (Reviews/ Perspectives only)	.doc, .docx, .tex	Box 1, Box 2, etc.	Article	Included in the main Article File in word-editable format, or uploaded as a separate Word or TeX file under the file type 'Article'.	N/A
Supplementary Information	.pdf	Supplementary Information	Supplemental Material	Any Supplementary Figures, Tables, Methods, Notes, Discussion and References must be provided in a single separate file in PDF format. We recommend limiting the size of your Supplementary Information file to 50MB. ** Please note that Supplementary Information cannot be changed after the paper has been accepted **	X
Supplementary Data	.csv, .xlsx, .txt, .zip, .cif	Supplementary Data 1, Supplementary Data 2, etc.	Data Sets	Any Supplementary Data files should be supplied separately and should be labelled as Supplementary Data 1, etc. Legends for these should be given in the Editorial Requests Table (and not in the main Supplementary Information file). We recommend limiting the size of each Supplementary file to 50MB.	X

				** Please note that Supplementary Information cannot be changed after the paper has been accepted **	
Supplementary Audio	.avi, .mp2, .wav, .mp3	Supplementary Audio 1, Supplementary Audio 2, etc.	Supplemental Material	Any Supplementary Audio files should be supplied separately and should be labelled as Supplementary Audio 1, etc. Legends for these should be given in the Editorial Requests Table (and not in the main Supplementary Information file). We recommend limiting the size of each Supplementary file to 50MB. ** Please note that Supplementary Information cannot be changed after the paper has been accepted **	N/A

Supplementary Movies	.mp4, .mpeg, .flv, .3gp, .m4v, .mts, .mxf, .mpg, .mov, .m2p, .gif, .wmv	Supplementary Movie 1, Supplementary Movie 2, etc.	Video	Any Supplementary Movie files should be supplied separately and should be labelled as Supplementary Movie 1, etc. Legends for these should be given in the Editorial Requests Table (and not in the main Supplementary Information file). We recommend limiting the size of each Supplementary file to 50MB. ** Please note that Supplementary Information cannot be changed after the paper has been accepted **	N/A
Supplementary Software	.zip	Supplementary Software 1, Supplementary Software 2, etc.	Supplemental Material	Any Supplementary Software files should be supplied separately and should be labelled as Supplementary Software 1, etc. Legends for these should be given in the Editorial Requests Table (and not in the main Supplementary Information file). Supplementary Software must be supplied as a ZIP file. We recommend limiting the size of each Supplementary file to 50MB. ** Please note that Supplementary Information cannot be changed after the paper has been accepted **	N/A

Life sciences reporting summary	.pdf	Reporting Summary	Supplemental Material	For life science manuscripts, a final version of the life sciences reporting summary. https://www.nature.com/documents/nr-reporting-summary.pdf The reporting summary will be published alongside your manuscript and therefore it needs to accurately represent your work. Please take a close look at the reporting summary and make sure that everything is completed correctly. In the section "Reporting for specific materials, systems and methods", you need to tick a box for each item, according to whether or not it applies, which will result in irrelevant sections becoming hidden. In the subsequent sections no box should be left blank or completed as N/A, including when the response is negative. Also, please make sure to include your name and date at the top of the document.	X
Solar cells reporting summary	.pdf	Solar cells reporting summary	Supplemental Material	For solar cell manuscripts, a final version of the solar cells reporting summary. https://www.nature.com/documents/nr-photovoltaic-reporting.pdf	N/A
Lasing reporting summary	.pdf	Lasing reporting summary	Supplemental Material	For lasing manuscripts, a final version of the lasing reporting summary. https://www.nature.com/documents/nr-lasing-reporting.pdf	N/A
Suggested feature image	.jpg, .pdf, .gif, .tiff, .psd	Featured image	Related Manuscript File	If you wish, an interesting image (but not an illustration or schematic) for consideration as a 'Featured Image' on the journal homepage. The file should be 1400x400 pixels in RGB format and should be uploaded as 'Related Manuscript File'. In addition	N/A

				to our home page, we may also use this image (with credit) in other journal-specific promotional material. If you submit a suggested featured image, please also include a completed image License to Publish form (also upload as 'Related Manuscript File', with file name 'Featured image LTP').	
--	--	--	--	--	--

COMMENTS TO THE EDITORS

“A COMPREHENSIVE UPDATED CROSS-SECTIONAL AND LONGITUDINAL META-ANALYSIS OF CYTOKINES IN EATING DISORDERS”

Johanna L. Keeler, Charlotte Bovenberg, Hubertus Himmerich,
Janet Treasure, Ben Carter, Ulrike Schmidt, Bethan Dalton

I thank the editors of *Communication Medicine* for inviting me to peer review the manuscript “*A comprehensive updated cross-sectional and longitudinal meta-analysis of cytokines in eating disorders.*” This assignment has allowed me to deepen my understanding of my primary area of interest, eating disorders. Despite their severe consequences and high mortality rates, eating disorders are still considered a niche specialty and are often overlooked by journals (Marzola et al., 2022). These challenges in research dissemination may have serious implications for clinical advancement (Marzola et al., 2022).

This study presents a comprehensive meta-analysis examining cytokine levels in individuals with eating disorders (EDs) compared to healthy controls, with particular attention to subtype differences and longitudinal changes across time points. A total of 43 studies were included. In anorexia nervosa (AN) compared to healthy controls (HCs), IL-6 and IL-15 levels were higher, while IL-7 levels were lower. No differences were found for TNF- α , IL-1 β , IL-4, IL-8, IL-10, IFN- γ , MCP, and TGF- β when controlling for outliers. Longitudinally, IL-6 levels were lower at follow-up compared to baseline in AN, while TNF- α and IL-1 β remained unchanged. No differences emerged based on AN subtype. Individuals with bulimia nervosa (BN) showed no differences in IL-6 and TNF- α compared to HCs. The available data were insufficient to conduct meta-analyses for binge-eating disorder (BED) or other EDs.

This is a well-conducted meta-analysis of a large number of studies, offering a robust overview of cytokine alterations in EDs. The statistical methods employed are both innovative and appropriate, enhancing the reliability and validity of the findings. The Introduction clearly outlines the need for this research and provides a comprehensive overview of immunology in EDs. The Discussion thoughtfully contextualizes the findings within the broader literature, offering a clear interpretation of their clinical and theoretical implications.

I suggest the following **minor points** for consideration, listed in order of appearance, as hints for improving the manuscript:

Abstract:

1. The full name of “IFN- γ ” should be provided when first mentioned.
2. The text states: “*Conclusion: In acute AN, concentrations of IL-6 and IL-15 are elevated and IL-7 is decreased, with evidence for normalisation of IL-6 over the course of weight restoration.*” The findings, as explained in the Results, show a reduction with a small effect size; however, there is no mention of “*normalisation*”. Accordance should be reached.
3. Keywords should be added at the end of the Abstract.

Materials and Methods:

4. Page 6: The adherence to the PRISMA checklist and the quality assessment should be mentioned in the Abstract, if word count allows.
5. Page 7, section “2.3. Source Selection”: The sentence “*The search process was conducted independently by two reviewers (JLK and CB)*” should be complemented with information on how discrepancies were resolved.

Results:

6. In Table 2, I recommend adding a column indicating whether the studies controlled for clinical and lifestyle confounders that may affect cytokine levels, specifying which variables were considered.
7. Page 20, end of section “3.1. Characteristics of included studies”: “CBT” appears twice, which may be unintended.
8. In the sections “3.3.1.1. Tumour-necrosis factor- α (TNF- α)” and “3.3.1.2. Interleukin-6 (IL-6)”, the number of participants included in the AN subtype analysis should be reported.
9. In the sections “3.3.1.4. Other cytokines available for meta-analysis” and “3.3.2. Bulimia nervosa”, information regarding the number of studies and total participants should be included.
10. In the section “3.4. Longitudinal meta-analyses in AN”, it is stated: “*As sensitivity analyses, studies were removed from analyses where either $\leq 10\%$ weight gain occurred, where participants didn’t reach at least 80% ideal body weight, or where the follow-up group mean BMI was $\leq 17\text{kg/m}^2$, depending on what data was reported in the study.*” Considering the global worsening of the clinical picture in cases of severe-enduring eating disorders (SE-ED) (Wonderlich et al., 2020), a complementary analysis focusing on time points without BMI improvement would have been valuable. If not feasible due to a lack of studies, this should

be acknowledged as a limitation of the current literature, suggesting future research in this direction.

11. In the sections “3.4.1. *Tumour necrosis factor- α (TNF- α)*”, “3.4.2. *Interleukin-6 (IL-6)*”, and “3.4.3. *Interleukin-1 β* ”, the time span between assessments should be reported. This information should also be addressed in the Discussion and Abstract. The same applies to the change in BMI between time points (Δ BMI), if available.

Discussion:

1. A further limitation of the existing literature – and a potential explanation for inconsistent findings – is that cytokine levels are influenced by circadian rhythms and may vary depending on the time of sampling (Nilsson et al., 2016; Zeng et al., 2024; Zielinski and Gibbons, 2022). I would suggest that the authors explicitly acknowledge this as a limitation, noting that the studies included did not control for time of blood collection. Additionally, I suggest recommending in Table 5 greater standardization on this point in future research.

References:

- Marzola, E., Panero, M., Longo, P., Martini, M., Fernández-Aranda, F., Kaye, W.H., Abbate-Daga, G., 2022. Research in eating disorders: the misunderstanding of supposing serious mental illnesses as a niche specialty. *Eating and Weight Disorders - Studies on Anorexia, Bulimia and Obesity* 27, 3005–3016. <https://doi.org/10.1007/s40519-022-01473-9>
- Nilsson, G., Lekander, M., Åkerstedt, T., Axelsson, J., Ingre, M., 2016. Diurnal Variation of Circulating Interleukin-6 in Humans: A Meta-Analysis. *PLoS One* 11, e0165799. <https://doi.org/10.1371/journal.pone.0165799>
- Wonderlich, S.A., Bulik, C.M., Schmidt, U., Steiger, H., Hoek, H.W., 2020. Severe and enduring anorexia nervosa: Update and observations about the current clinical reality. *International Journal of Eating Disorders* 53, 1303–1312. <https://doi.org/10.1002/eat.23283>
- Zeng, Y., Guo, Z., Wu, M., Chen, F., Chen, L., 2024. Circadian rhythm regulates the function of immune cells and participates in the development of tumors. *Cell Death Discov* 10, 199. <https://doi.org/10.1038/s41420-024-01960-1>
- Zielinski, M.R., Gibbons, A.J., 2022. Neuroinflammation, Sleep, and Circadian Rhythms. *Front Cell Infect Microbiol* 12. <https://doi.org/10.3389/fcimb.2022.853096>